# An enzyme activation network reveals extensive regulatory crosstalk between metabolic pathways

Sultana Mohammed Al Zubaidi[1,6], Muhammad Ibtisam Nasar [ID][1,6], Richard A Notebaart[2], Markus Ralser [ID][3,4,5] & Mohammad Tauqeer Alam [ID][1✉]

## Abstract

**Enzyme activation by cellular metabolites plays a pivotal role in regulating metabolic processes. Nevertheless, our comprehension of such activation events on a global network scale remains incomplete. In this study, we conducted a comprehensive investigation into the optimization of cell-intrinsic activation interactions using *Saccharomyces cerevisiae* metabolic network as the basis of the analysis. To achieve this, we integrated a genome-scale metabolic model with cross-species enzyme kinetic data sourced from the BRENDA database, and to use this model as a basis to estimate the distribution of enzyme activators throughout the cellular network. Our findings indicate that the vast majority of biochemical pathways encompass enzyme activators, frequently originating from disparate pathways, thus revealing extensive regulatory crosstalk between metabolic pathways. Notably, activators have short pathway lengths, indicating they are activated quickly upon nutrient shifts, and in most instances, these activators target key enzymatic reactions to facilitate downstream metabolic processes. Interestingly, highly activated enzymes are substantially enriched with non-essential enzymes compared to their essential counterparts. This observation suggests that cells employ enzyme activators to finely regulate secondary metabolic pathways that are only required under specific conditions. Conversely, the activator metabolites themselves are more likely to be essential components, and their activation levels surpass those of non-essential activators. In summary, our study unveils the widespread importance of enzymatic activators and suggests that feed-forward activation of conditional metabolic pathways through essential metabolites mediates metabolic plasticity.**

**Keywords** Intracellular Activation Network; Metabolic Regulation; Regulatory Crosstalk; Metabolic Network; Activator Compounds
**Subject Categories** Metabolism; Microbiology, Virology & Host Pathogen Interaction

## Introduction

The metabolic network, the largest interconnected system of the cell (Jeong et al, 2000; Mahadevan and Palsson, 2005; Nielsen, 2017), is an essential biological system by which organisms produce necessary components such as energy molecules and cellular building blocks including carbohydrates, lipids, nucleotides, proteins, and vitamins for their growth and survival (Nielsen, 2017). To survive in changing environments, the regulation of cellular metabolism through tuning enzyme abundance or activities is essential (Klosik et al, 2017). Cells achieve metabolic regulation by a range of regulatory interactions (Metallo and Vander Heiden, 2013; Zhu and Thompson, 2019), which occur at different levels of cellular processes. These include hierarchical control, which includes transcriptional and translational level regulations to change the enzyme level, the posttranslational modification of enzymes to change their activity, and metabolite-enzyme interactions that can inhibit or activate enzymes (Chubukov et al, 2013; Daran-Lapujade et al, 2007; González-Arrué et al, 2023). As the latter is driven by cell-intrinsic metabolites that are formed within or outside the regulated pathway, the metabolic network is in part self-regulatory (Basler et al, 2016; Grüning et al, 2010; Reznik et al, 2017). Characterizing these metabolite-enzyme interactions is thus key to understanding metabolic phenotypes.

Broadly speaking, enzyme inhibitory regulations can be allosteric, non-competitive, uncompetitive, and competitive. Among these, the most frequent are the competitive interactions, in which the inhibitor competes with the substrates to bind to the enzyme's active site (Alam et al, 2017), reducing the rate of enzymatic reactions (Buker et al, 2019). Competitive inhibitors are often the by-product of the same biochemical pathway or created in close proximity within the metabolic network (Alam et al, 2017). Indeed, enzyme inhibitory interactions in human metabolism emerge merely due to a finite chemical diversity that prevails within the metabolome (Alam et al, 2017). In contrast, enzyme activation interactions enhance the rate of metabolic reactions, and most activators bind allosterically (Enzymes as Drug Targets, 2017). For example, Phosphofructokinase-1 (PFK-1, EC: 2.7.1.11), one of the most important glycolysis regulatory enzymes, is allosterically activated by a number of cellular metabolites including ADP, GDP,

[1]Department of Biology, College of Science, United Arab Emirates University, Al Ain, Abu Dhabi, UAE. [2]Food Microbiology, Wageningen University and Research, Wageningen, The Netherlands. [3]Department of Biochemistry, Charité – Universitätsmedizin Berlin, Berlin, Germany. [4]The Centre for Human Genetics, Nuffield Department of Medicine, Oxford, UK. [5]Max Planck Institute for Molecular Genetics, Berlin, Germany. [6]These authors contributed equally: Sultana Mohammed Al Zubaidi, Muhammad Ibtisam Nasar.
✉E-mail: mtalam@uaeu.ac.ae

AMP, PI, Glutathione, but the most potent activator of PFK-1 is fructose 2,6-bisphosphate (F-2,6-BP) that is produced by Phosphofructokinase-2 enzyme (PFK-2, EC: 2.7.1.11). While individual examples have been well studied, the extent of activation interactions is not entirely characterized at a cellular-wide level.

Non-catalytic enzyme–metabolite interactions remain challenging to study, with the consequence that we still lack a systems- or network-wide understanding of the role of intrinsic small molecule enzyme activators. To overcome this limitation to learn about the general principles of enzyme activation across pathways, we have herein reconstructed a network of enzyme–metabolite activatory interactions by enriching the genome-scale metabolic network of *Saccharomyces cerevisiae* (Data ref: Zhang et al, 2023) with cross-species enzyme activation data from the Braunschweig Enzyme Database (BRENDA) database, which has collected enzyme kinetic data over a century of biochemical research (Data ref: Chang et al, 2021). While this cross-species approach certainly has limitations, as not all activators are necessarily conserved across species, it nevertheless provided valuable insights into the general properties of enzyme activation. The cell-intrinsic enzyme–metabolite activation network indicates that enzyme activation is extremely frequent, and that up to 54% of enzymatic reactions could be intracellularly activated. Activated enzymes are found across the entire metabolic system, covering most of the biochemical pathways. Moreover, in contrast to enzyme–metabolite inhibitory interactions, where most pathways predominantly self-inhibit, enzyme–metabolite activation interactions primarily exhibit trans-activation between pathways. We show that while most of the highly activated enzymes are non-essential, the highly activating compounds are essential for growth and are often produced shortly after nutrient uptake. In addition, we compared the enzyme–metabolite activation interaction with the genetic interaction and examined the co-expression pattern of the genes associated with activating metabolites and activated enzymes by analyzing transcriptome profiles from more than 600 unique conditions from >200 experiments and proteome profile of hundreds of knockout strains (Data ref: Messner et al, 2023; Data ref: Parkinson et al, 2007). We conclude that the cell-intrinsic metabolic activation interactions could be used to understand the observed metabolic profile in various conditions.

# Results

## Cell-intrinsic activation of the metabolic network

Despite the fact that not all metabolic reactions are conserved across all species, in the previous two decades, the aggregation of cross-species information on metabolic reactions allowed the successful reconstruction of genome-scale metabolic networks (Edwards and Palsson, 2000; Förster et al, 2003). Later, the same principle also revealed general properties of metabolic regulation and inhibitory interactions between metabolic pathways (Alam et al, 2017). Herein, we used this strategy to gain network-scale insights into enzyme activators.

To construct a genome-scale enzyme–metabolite activation interactions network, we obtained the topology of the *Saccharomyces cerevisiae* metabolic network and enriched it with cross-species acquired data of enzyme activation. For each metabolic enzyme

(in total 635 enzymes) in the genome-scale metabolic model (Yeast9) (Data ref: Zhang et al, 2023) a list of all associated activatory molecules were downloaded from the Brenda database, using SOAP clients instructions (Data ref: Chang et al, 2021). To justify the use of cross-species data, we examined how many of these 635 *S. cerevisiae* enzymes had activation evidence from both S. cerevisiae and other species in the BRENDA database. Among them, we identified kinetic information for 140 *S. cerevisiae*-specific enzymes. Notably, 88 of these enzymes (63%) shared at least one activation interaction with the same activator molecule across different species. This high level of concordance supports our assumption that activation interactions are conserved across species.

The obtained list of activators includes various molecules, ranging from non-cellular compounds like drugs to intracellular metabolites produced within the cell. Enzyme assays confirm that all these metabolites enhance the enzyme's activity. By comparing the list of activatory molecules with the intracellular metabolites of the metabolic model (Zhang et al, 2023) all non-cellular molecules were removed (Fig. 1A; see "Methods" for details) to produce the cell-intrinsic activation interactions network (Fig. 1B; Dataset EV1). In the network, the nodes represent enzymes and activator metabolites, and the edges are formed between nodes when an enzyme is activated by a metabolite (Fig. 1B). The final network comprises 1499 activatory interactions involving 344 enzymes and 286 cellular metabolites. While these enzymes and metabolites are derived from the Yeast9 model, the activation interaction evidence is based on cross-species data. Like most other biological networks, the cell-intrinsic activation interactions network is scale-free and follows the power law distribution (Fig. EV1). The degree of activation for enzyme or activator metabolite nodes corresponds to the number of activating interactions, represented by the count of edges connected to each node.

In terms of the metabolic network coverage, 54% of the total metabolic enzymes (344 out of 635) are intracellularly activated, and the remaining 46% of the metabolic enzymes are either activated by extracellular molecules (121 enzymes out of 635; 19%) or contain no activation interactions at all (170 enzymes out of 635; 27%) (Fig. 1C). On the other hand, out of the total 1378 metabolites of the Yeast9 metabolic model (Data ref: Zhang et al, 2023), 20.7% (286 in total) act as activators on at least one enzyme (Fig. 1D). To understand the prevalence of activators within different compound classes, all cellular metabolites were clustered into 8 different compound groups based on the KEGG database classification (Kanehisa et al, 2016) (Fig. 1E). Lipids, which are highly prevalent among inhibitory metabolites (Alam et al, 2017), have a low prevalence of activatory metabolites. They are the only metabolite group, besides the unclassified "Others", that has fewer activators than non-activators, and they are associated with even fewer activatory interactions (Fig. 1E). On the other hand, metabolites belonging to "Nucleosides, Nucleotides, and Analogs", "Amino Acids, Peptides, and Analogs", "Carbohydrates and Carbohydrate Conjugates", "Aliphatic Acyclic Compounds", and "Organic Acids and Derivatives" have substantially higher prevalence of activatory metabolites compared to non-activator metabolites (Fig. 1E). Besides, the maximum percentage of activatory interactions are conducted by the 'Nucleosides, Nucleotides, and Analogs' (~19%), followed by "Amino Acids, Peptides, and Analogs" (~16.6%), which explains the important roles of these metabolite groups in positively regulating metabolic reactions (Fig. 1E).

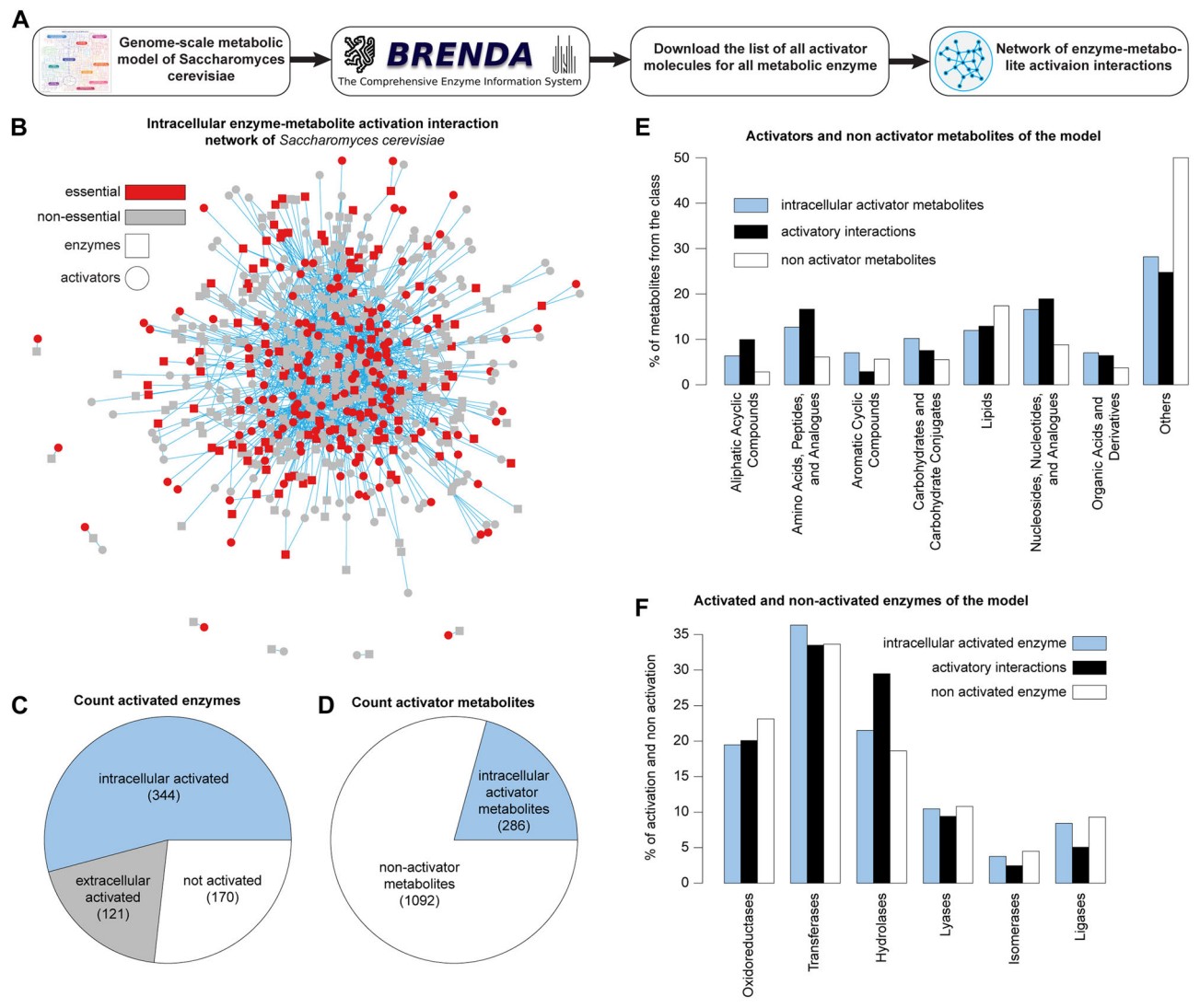

**Figure 1. Cell-intrinsic enzyme–metabolite activation interactions network.**

(A) Schematic of reconstructing cell-intrinsic enzyme–metabolite activation interactions network. (B) The network of enzyme–metabolite activation interactions within *Saccharomyces cerevisiae* contains 344 enzymes and 286 metabolites as nodes, and they are connected by 1499 links representing enzyme–metabolite activation. The essential and non-essential components are marked in red and gray, respectively. (C) Analysis of the distribution of enzymes across metabolism, categorizing them as intracellularly activated, extracellularly activated, or unactivated. (D) Proportion of activator and non-activator metabolites within the metabolic network. (E) The prevalence of intracellular activator metabolites, non-activator metabolites, and activatory interactions within different compound categories. (F) The prevalence of intracellular activated enzymes, activatory interactions, and non-activated enzymes within different enzyme classes.

In the network of human enzyme inhibition interactions (Alam et al, 2017) all enzyme classes were equally susceptible to metabolic inhibitions. In contrast, in the activation network (Fig. 1B) each enzyme class has a different prevalence of activated enzymes (Fig. 1F). Out of the total intracellular activated enzymes, almost 36.33% belong to Transferases that is also the largest enzyme class of the Yeast9 metabolic network (Data ref: Zhang et al, 2023), and are equally associated with all cell-intrinsic activated interactions (33.48%) (Fig. 1F). Isomerases, and Ligases, which are the two smallest enzyme classes of the metabolic network, have even lower ratios of activated enzymes compared to non-activated enzymes (3.77% and 8.43%, respectively) (Fig. 1F). In contrast, Oxidoreductases and Hydrolases, which are 2nd and 3rd largest metabolic enzyme classes, catalyzing thermodynamically favored metabolic

reactions (nonequilibrium), have substantially higher percentage of intracellularly activated enzymes (19.47% and 21.51%, respectively, Fig. 1F).

## Activatory interactions are distributed across metabolism

This strategy mapped an intracellular metabolic activator to 54% of all metabolic enzymes (Fig. 1C). Enzyme activation can thus occur throughout metabolism, covering the majority of the metabolic pathways that have more than 2 enzymatic reactions (65 out of 67 Yeast9 pathways; 97%) (Fig. 2A). Some of the highly intracellularly activated metabolic pathways were Pyruvate metabolism (18/28), Glycolysis/Gluconeogenesis (17/24), Purine and Pyrimidine

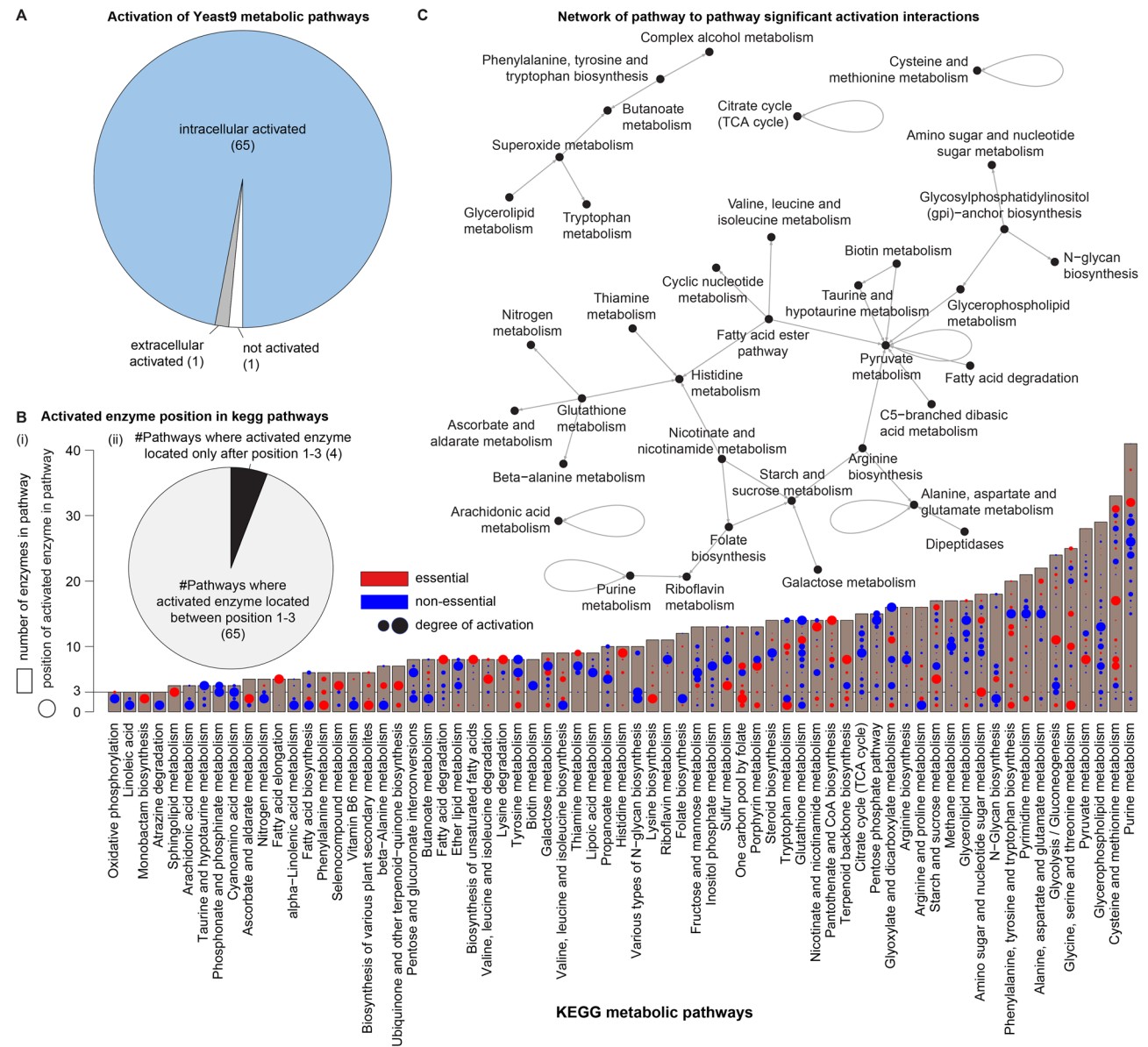

**Figure 2. Activation of metabolic pathways.**

**(A)** A pie chart depicting the number of pathways associated with intracellularly activated, extracellularly activated, and non-activated enzymes shows that nearly every biochemical pathway exhibits activatory interactions (65 out of 67). **(B)** (i) Activated enzymes, whether essential or non-essential with varying degrees of activation, are distributed in various positions within metabolic pathways. (ii) Remarkably, most biochemical pathways (94%) include at least one enzymatic reaction within the initial three positions of the pathway. **(C)** The interconnected network of pathway-to-pathway activations highlights transactivation, where metabolites from one pathway significantly stimulate enzymes in other pathways. This is in stark contrast to the inhibitory interactions network (Alam et al, 2017), where the majority of pathways primarily inhibit their own enzymes. Enzyme localization in metabolic pathways is based on KEGG pathway definitions.

metabolism (20/41 and 11/21, respectively), and various amino acid metabolic pathways such as alanine, aspartate and glutamate metabolism, glycine, serine and threonine metabolism, cysteine and methionine metabolism, arginine biosynthesis, phenylalanine, tyrosine and tryptophan metabolism, and tryptophan metabolism (Fig. 2Bi). Moreover, the enzymes of some of these pathways are among the most abundant and most regulated enzymes also in an unbiased proteome investigation (Messner et al, 2023). Furthermore, we also observe that some of the metabolic pathways are enriched with mostly essential

activated enzymes (e.g., Lysine biosynthesis, Phenylalanine metabolism, Histidine metabolism, Pantothenate and CA biosynthesis, and Phenylalanine, tyrosine and tryptophan biosynthesis), whereas in some pathways none of the activated enzymes seems essential for growth (for example, Biotin metabolism, Lipoic acid metabolism, Taurine and hypotaurine metabolism, Various types of N-glycan biosynthesis, and Citrate cycle (TCA cycle)).

Furthermore, we examined the position of these activated enzymes in "KEGG metabolic pathways". We find that at least one of the first

three initial enzymatic reactions of most of the pathways in KEGG (65/69; 94.2%) are positively regulated (Fig. 2Bii). In biochemical pathways, reactions at the beginning are generally nonequilibrium flux-generating reactions which must be regulated to change the flux through the entire pathway (Moreno-Sánchez et al, 2008). In addition, we also observed that the activated enzymes are not only restricted to the beginning of the biochemical pathways but they are also distributed throughout the pathways. This spread of activated enzymes shows that the other distant reactions could also be regulated by intracellular metabolites, depending on conditions. However, on the other hand, we observe that out of 235 enzymes, which are located after position 3 in various chemical pathways, 64 enzymes (27%) are also placed between position 1–3 in other pathways. Further, it is important to note that the average length of pathways (number of enzymes) is 12.36, and 47.8% of the pathways have more than 10 enzymes and 81% of pathways have more than 5 enzymes. Lastly, we did not observe any particular pattern of enzyme essentiality or the degree of activation for early or late regulated enzymes of the pathway (Fig. 2Bii).

## Evidence for a high degree of transactivation within the metabolic network

Next, we examined the extent to which different biochemical pathways activate one another (Fig. 2C). Among the 72 metabolic pathways in the Yeast9 network, half (36 or 50%) exhibit significant activation interactions with other pathways ($P$ value < 0.05, hypergeometric test). These pathways either activate, are activated by, or both activate and are activated by other pathways. The remaining pathways, despite containing activated enzymes, do not show significant activation interactions with other pathways ($P$ value < 0.05, hypergeometric test). Among the 36 pathways with activation interactions, the majority (26, or 72%) engage in only one type of interaction—either being activated by other pathways without activating any or activating other pathways without being activated themselves. In addition, we observe that 13 pathways activate only a single pathway, 17 are activated by just one pathway, and only 4 pathways exhibit significant self-activation. This pattern highlights the general specificity of activation interactions within the network. This notable degree of transactivation between pathways reflects the interconnected nature of cellular metabolism. It ensures that the metabolic pathways are specifically activated and coordinated to dynamically adapt in order to meet the cellular needs.

In addition, only six pathways are activated by more than one pathway, revealed the enrichment analysis ($P$ value < 0.05, hypergeometric test). One notable pathway that is significantly activated by the largest number of pathways (eight in total) is pyruvate metabolism. This pathway is known to be highly regulated because it serves as a key metabolic hub, linking glucose breakdown to various cellular processes. Pyruvate metabolism acts as a bridge between glycolysis and several major biochemical processes, including the citric acid cycle, anaerobic fermentation, and anabolic pathways for synthesizing amino acids, depending on cellular needs. Similarly, due to the broad role of Histidine metabolism in protein synthesis, energy production, nitrogen regulation, and stress response, and its dynamic regulation according to cellular need, it is also activated by many specific pathways. There are some pathways including Fatty acid ester pathway, Glutathione metabolism, Arginine biosynthesis, Glycosylphosphatidylinositol (GPI)-anchor biosynthesis, and Nicotinate and nicotinamide metabolism which act as hub activators, activating more than one pathway significantly.

## Highly activating metabolites are mostly essential, whereas highly activated enzymes are mostly non-essential for growth

The essentiality of enzymes and metabolites, predicted using the flux balance analysis approach for optimal growth in glucose minimal media, was integrated with the cell-intrinsic activation network (Fig. 1B). An enzyme or metabolite is deemed essential if the Yeast9 metabolic model fails to produce biomass in glucose minimal media (standard laboratory condition) after all reactions associated with that enzyme or metabolite are removed. Otherwise, the enzyme or metabolite is considered non-essential, though they may be conditionally essential under different conditions. We observe a significant difference in the prevalence of essentiality of activators compared to non-activator metabolites ($P$ value = 1.99e–38, Fisher's Exact Test). Nearly half of the cellular activator metabolites are essential (45%) for growth, whereas the essentiality ratio is merely 11.2% for non-activators (Fig. 3A). In contrast, activated and non-activated enzymes have no significant difference in the prevalence of essential enzymes ($P$ value = 0.15, Fisher's Exact Test), although non-activated enzymes are slightly less prevalent with essential enzymes (30%) compared to activated enzymes (34%) (Fig. 3A).

We analyzed the degree of activation for both essential and non-essential metabolites and found that 77% (17 out of 22) of hub activator metabolites, which activate more than 15 enzymes (degree >15), are essential (Fig. 3Bi,ii). In concurrence, the number of interactions associated with essential metabolites is significantly higher compared to non-essential activators ($P$ value = 8.28e − 05, $T$ test), where 45% of the essential activatory metabolites are activating more than 64% of the interactions in the network (Fig. 3Biii). In contrast, the highly activated enzymes appear to be non-essential for growth. We found that 73.3% (11 out of 15) of hub enzymes, which are activated by more than 15 metabolites (degree >15), are non-essential for growth (Fig. 3Ci,ii). Although the overall activation degree for non-essential enzymes is not significantly different from that of essential enzymes ($P$ value = 0.128, $T$ test) (Fig. 3Ciii), this insignificant difference is due to the large number of enzymes with a degree of 1. Moreover, examining the highly connected hub nodes [activators and enzymes] with different numbers of connections (12, 15, and 20) give the same conclusions (Fig. EV2A,B).

It has previously been reported that essential enzymes are tightly regulated to maintain constant gene expression levels regardless of genetic or environmental perturbations (Grüning et al, 2010; Yang et al, 2012). Consequently, these genes are expected to exhibit less variation in gene expression (Park and Lehner, 2013). Consistent with those studies (Grüning et al, 2010; Park and Lehner, 2013; Yang et al, 2012), using microarray gene expression data across hundreds of conditions, we find that essential enzymes (whether activated or non-activated) have higher gene expression levels compared to non-essential enzymes, and effect size is small but having meaningful difference (median log2 gene expression 10.78 vs. 10.11, SD 1.66 vs 1.92, Cohen's $d = 0.303$, $P$ value < 2.2e−16 $T$ test, Fig. EV3A). We did not observe any substantial and meaningful difference in the gene expression levels between

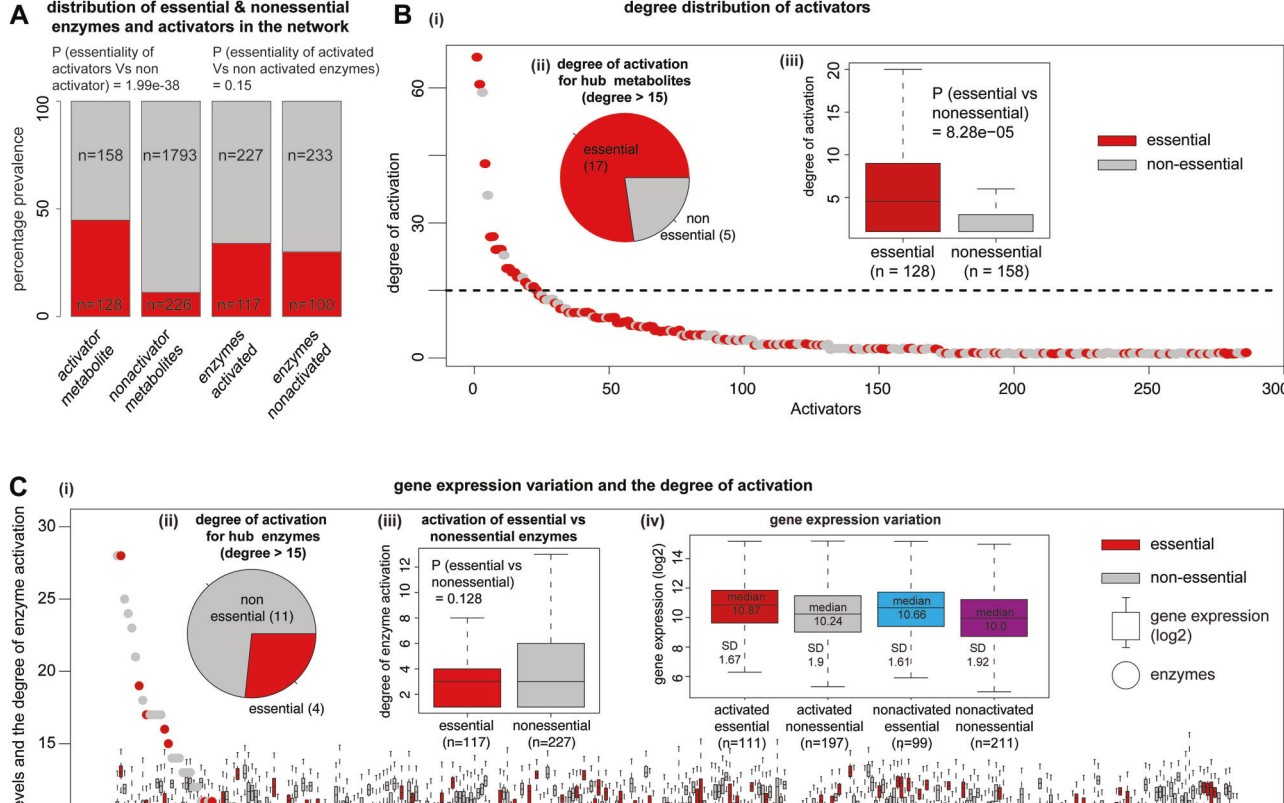

**Figure 3. Degree distribution of activators and enzymes.**

(A) In contrast to non-activator metabolites, activators demonstrate a significant increase in essential metabolites (*P* value = 1.99e-38, Fisher's Exact Test). However, when it comes to enzymes, there are no significant differences in essentiality levels between activated and non-activated enzymes (*P* value = 0.15, Fisher's Exact Test). (B) The degree distribution of activator metabolites within the cell-intrinsic activation interaction network reveals that highly interactive activators are mostly essential for growth (i, ii). Furthermore, metabolites essential for growth (*n* = 128) show significantly more activatory interactions compared to non-essential activatory metabolites (*n* = 158) (*P* value = 8.28e − 05, *t* test) (iii). (C) The variability in gene expression across over 500 conditions for all activated enzymes, whether essential or non-essential, was assessed in relation to their degree of interactions (i). The distribution of degrees among activated enzymes in the cell-intrinsic activation interaction network indicates that highly activated hub enzymes (degree >15) are not essential (i, ii). However, considering all activated enzymes, there are no significant differences in the degree of activation between essential (*n* = 117) and non-essential enzymes (*n* = 227) (*P* value = 0.128, *t* test) (iii). In addition, different levels of expression are evident in the global gene expression profiles among activated essential enzymes, activated non-essential enzymes, non-activated essential enzymes, and non-activated non-essential enzymes where activated essential enzymes exhibit the highest median gene expression value (iv). Box plots display the median, the upper and lower quartiles, and the minimum and maximum values represented by the whiskers.

activated and non-activated enzymes, regardless of their essentiality (median log2 gene expression 10.47 vs. 10.22, SD 1.84 vs 1.86, Cohen's *d* = 0.12, *P* value < 2.2e−16 *T* test, Fig. EV3B). However, enzymes that are essential for growth and are intracellularly activated exhibit the highest median gene expression values and the least variance in contrast to enzymes that are non-essential for growth and rely on activation for their expression (median log2 gene expression 10.87 vs 10.0, SD 1.67 vs 1.92, Cohen's *d* = 0.4 between the two groups, *P* value < 2.2e−16 *T* test, Fig. 3Civ). This indicates that these essential activated enzymes are consistently upregulated and maintain the highest expression levels in comparison with non-essential non-activated enzymes. The

Cohen's *d* that measure of effect size between essential plus activated and non-essential plus non-activated is 0.4, suggesting a meaningful difference between these groups. We further conducted a pathway-specific analysis to assess gene expression variation between essential and non-essential enzymes. In line with the expression profile of overall enzymes, in the majority of pathways (34 out of 50), genes linked to essential enzymes demonstrate a higher median gene expression compared to those associated with non-essential enzymes (Fig. EV3C). Similarly, we examined protein expression variations among essential, non-essential, activated, and non-activated enzymes using proteome profiles from thousands of knockout strains, as measured by liquid

chromatography tandem mass spectrometry (Messner et al, 2023). Our findings show consistent expression patterns in both proteome and transcriptome datasets (Fig. EV4). It is important to note that we are not studying metabolic regulation through mRNA or protein expression; rather, we are using expression variation data to detect differences between essential and non-essential enzymes, as well as activated and non-activated enzymes, since approximately 50% of the variation in metabolite concentrations can be attributed to changes in enzyme abundance (Zelezniak et al, 2018).

## Activatory metabolites are produced shortly after nutrient uptake

We have analyzed the metabolic network topology and examined how activator and non-activator metabolites are produced within the metabolic network (Data ref: Zhang et al, 2023). First, we converted the metabolic network model into a graph where nodes represent metabolites and edges are established when two metabolites are connected by a chemical reaction. We removed 25 highly connected metabolites (metabolites, mostly cofactors, with more than 50 metabolic connections in any cellular compartment; Dataset EV1) in order to avoid misleading connections. Using the simplified network, we calculated the shortest path length, representing the minimum number of chemical transformations required to produce different activators and non-activator metabolites from the glucose metabolite, the main carbon source nutrient. We find that within the metabolic network the shortest path length from glucose to activators is significantly lower compared to non-activators metabolites (Fig. 4A, $P$ value = 9.52e−38, $T$ test). The bulk of activator metabolites exhibit a shortest path length of almost 6, contrasting with non-activators which is almost 8 (with mean, median and standard deviation of shortest path lengths for activators at 6.29, 6, and 1.71, respectively, and for non-activators at 7.99, 8, and 1.54, Fig. 4A,B). Moreover, we note that activators positioned 6 connections away from glucose are associated with substantially higher number of activatory interactions compared to those adjacent to glucose or distantly located within the metabolic network (Fig. 4C). Furthermore, we also find a significant difference between the shortest path length considering all exchange nutrient metabolites to activators compared to non-activators (Fig. EV5). These results suggest an inherent optimality in the production of cellular activator metabolites within the metabolic network. This indicates that intracellular activators begin to form shortly after nutrient uptake, subsequently driving overall metabolism in a positive direction. Furthermore, this optimality is not unique to activators but represents a fundamental feature of metabolic systems. This is demonstrated by the finding that only 12 key precursor metabolites, located at various branch point nodes and converted from carbon source nutrients via the shortest path lengths, form the foundation of cellular biomass (Noor et al, 2010).

In contrast, we did not observe any major difference between activated and non-activated enzymes in terms of the shortest path length, considering glucose exchange reaction as the root (Fig. 4D,E). For the majority of activated and non-activated enzymes, the shortest path length is 4–5 (with mean, median, and standard deviation of shortest path lengths for activated enzyme at 4.3, 4, and 1.24, respectively, and for non-activators at 4.67, 5, and 1.30; Fig. 4D,E). Interestingly, we note that enzymatic reactions situated within the shortest path length of 3–5 from glucose exchange reactions demonstrate the highest number of activatory

interactions compared to enzymes positioned further away (>5 shortest path length) from glucose exchange reactions (Fig. 4F).

## Co-expression of genes associated with activatory interactions

We have analyzed the co-expression pattern of genes linked with nodes of every interaction of the cell-intrinsic activation network. For this analysis, highly connected hub metabolites such as Phosphate, AMP, ADP, and ATP molecules were removed due to their connection with hundreds of genes. Genes linked to the metabolic reaction for producing activatory metabolites and activated enzymes were based on the Yeast9 metabolic model definition (Data ref: Zhang et al, 2023). The gene expression data from 207 experiments consisting of 665 unique conditions were obtained from the ArrayExpress database (microarray experimental data from Genome2 chips) and processed using the limma package in R to produce the gene expression profile for every gene (please see the method section for details).

For each pair of genes, corresponding to the nodes of the cell-intrinsic activation interaction, we have performed the correlation test, and considered the genes co-expressed based on a stringent threshold cutoff ($r > 0.75$ and adjusted $P$ value <0.001, correlation test). Figure 5A shows a subnetwork of the main cell-intrinsic activation interaction in which genes associated with both nodes are co-expressed. Out of 1344 cellular activatory interactions (after removing cofactors from the main network), only 448 interactions (33.33%), produced from 100 metabolites and 148 enzymes, have co-expressed gene expression across conditions (Fig. 5A,B), which highlight that such interactions are quite common in constraining the metabolism in different conditions. Similarly, by comparing activatory interactions with genetic interaction networks, we find that 31.6% of the interactions are also genetically interacting, where genes associated with such interactions are genetically linked.

# Discussion

The metabolic network possesses inherent regulatory mechanisms to autonomously control or fine-tune its functionality by modulating enzyme activity through various enzyme–metabolite interactions (Chakrabarti et al, 2013; Herrgård et al, 2006; Khodayari and Maranas, 2016; Wang et al, 2017). These interactions can either be inhibitory, where the metabolite diminishes the enzyme's activity, or activatory, where cellular metabolites enhance the enzyme's activity (Alam et al, 2017; Kim and Copley, 2012; Lopina, 2017; Mahadevan and Palsson, 2005; Wang et al, 2017). In the metabolic system, numerous enzyme–metabolite inhibitory interactions exist, acting as constraints on metabolism. Studies indicate that over 75% of inhibitory interactions in humans are competitive, wherein structurally similar substrates and metabolites compete for the same active site. Consequently, inhibitory metabolites are frequently produced within the same pathway and significantly impede their own enzymatic reactions (Alam et al, 2017). However, comprehensive global characterization of activation interactions remains incomplete.

In this study, we systematically examined the cell-intrinsic activation interactions using cross-species data of enzyme activators collected by BRENDA (Data ref: Chang et al, 2021), projected on the

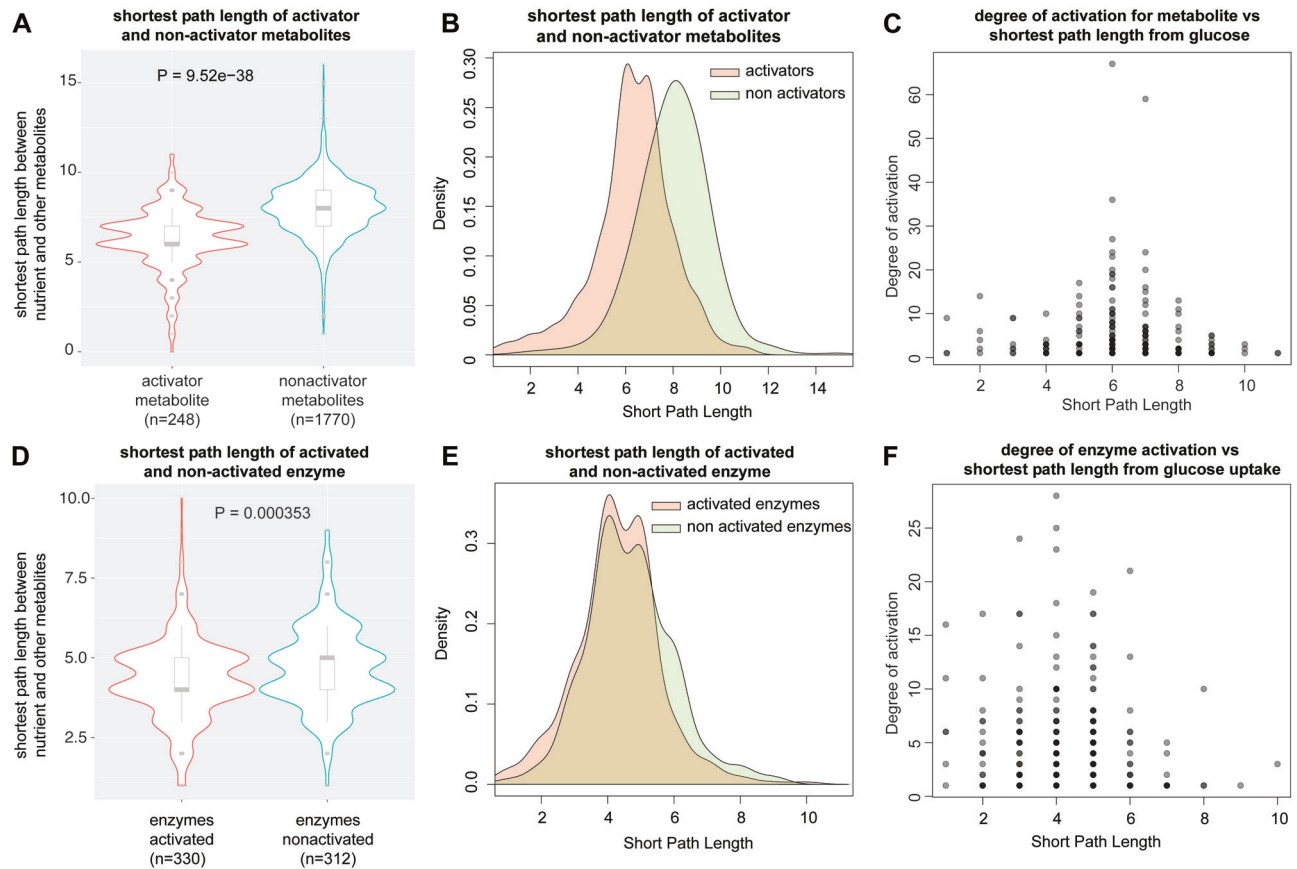

**Figure 4. Shortest path of activator metabolites and activated enzymes in the metabolic network.**

(A) Within the organism's metabolic network, the shortest path length was computed for all metabolites from glucose, the reference nutrient metabolite. Activator metabolites exhibit a significantly shorter shortest path length compared to non-activators ($P$ value $= 9.52e − 38$, $t$ test). (B) In addition, a majority of activators have a shortest path length of 5, while for non-activators, it is 6–7. (C) Analyzing the correlation between the shortest path length and the total number of activations from metabolites reveals a positive association between these two variables. (D) Likewise, the shortest path length was determined for all enzymes originating from the glucose transporter, serving as the reference nutrient uptake enzyme. While a statistically significant distinction exists between activated and non-activated enzymes concerning their shortest path length from the glucose transporter ($P$ value $= P = 0.000353$, $t$ test) (D), this disparity is primarily attributed to a minimal fraction of non-activated enzymes with a slightly higher shortest path length (that is 5). Beyond this, no variation is observed in the shortest path length between activated and non-activated enzymes (E). Finally, there is no correlation between the degree of enzyme activation and their respective shortest path lengths (F). Box plots display the median, the upper and lower quartiles, and the minimum and maximum values represented by the whiskers.

structure of the *Saccharomyces cerevisiae* metabolic network Yeast9 (Data ref: Zhang et al, 2023) that in itself has been derived from cross-species information (Edwards and Palsson, 2000; Förster et al, 2003). Unlike most inhibitory interactions (Alam et al, 2017), the enhancement of enzyme activity in this context necessitates allosteric binding of metabolites, facilitating the substrate binding to the active site (Tzeng and Kalodimos, 2009). The reconstructed cellular intrinsic activation network comprises 344 enzymes and 286 metabolites, interconnected by 1499 activatory interactions (Fig. 1B). It is important to note that our approach operates under the fundamental assumption that metabolic activatory interactions are conserved across species (Guan et al, 2020; Ribeiro et al, 2020). This is a necessary assumption for our study to achieve sufficient network coverage; no activator network is currently sufficiently mapped at the single-species level. While this assumption certainly has limitations (for example, our approach might overestimate the total number of activated interactions by an unknown degree), there is indeed evidence that many activation reactions are indeed conserved. For example, 63% of *S.*

*cerevisiae* metabolic enzymes share at least one activation interaction with other species. In addition, enzyme regulations in BRENDA and the literature do not provide enough information to determine the extent of their condition dependency, limiting our network's ability to achieve condition-specific resolution.

Initially, we investigated the extent to which cellular metabolites activate enzymatic metabolic reactions within the metabolic system and their distribution across various biological processes. In the cross-species network, 54% of the total metabolic enzymes are activated by other cellular metabolites (Fig. 1C). Among the remaining enzymes, approximately 19% are reported to be activated by external metabolites, while for 27% of the enzymatic reactions thus far no metabolic activator was reported (Fig. 1C). Notably, the activated enzymes are distributed throughout the metabolic network, encompassing almost all biochemical pathways (Fig. 2A). This comprehensive coverage of metabolic pathways with intracellularly activated enzymes elucidates the optimality of inherent cell-intrinsic activation constraints on metabolism, underscoring their crucial role in the

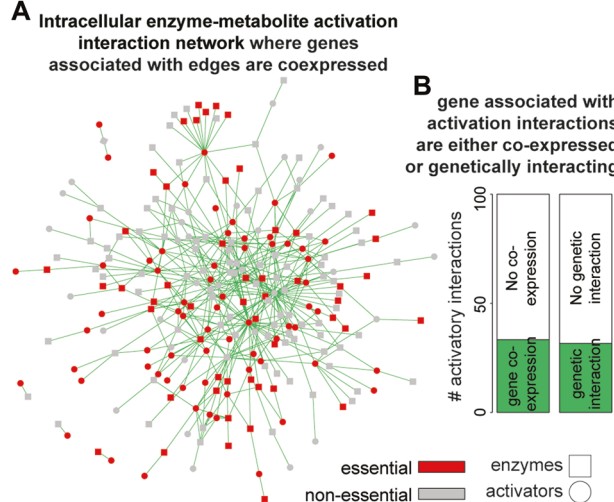

**A** **Intracellular enzyme-metabolite activation interaction network** where genes associated with edges are coexpressed

**B** gene associated with activation interactions are either co-expressed or genetically interacting

essential ■ enzymes □
non-essential ■ activators ○

**Figure 5. The global co-expression profile of genes associated with activatory interactions.**

(A) A portion of the activation network exhibiting co-expression of genes linked to both enzyme and metabolite nodes in the connection. The subnetwork highlights both essential and non-essential metabolites and enzymes.
(B) Within the activation network, approximately 33.3% of the total connections display notable co-expression between genes linked to activator production and the activated enzymes, as evidenced by a correlation coefficient $r > 0.7$ and $P$ value $< 0.01$. Similarly, concerning the global genetic interactions in *S. cerevisiae*, nearly 31.6% of interactions involve both essential genes.

efficient operation of metabolic pathways (Metallo and Vander Heiden, 2013). In addition, prior studies employing metabolic control analysis and flux coupling analysis support the observation that metabolic controls are widely dispersed across the metabolic network (Basler et al, 2016; Millard et al, 2017).

Moreover, our data suggest that in the majority of biochemical pathways, at least one of the top three enzymatic reactions can be activated intracellularly (Fig. 2Bi,ii). This underscores the remarkable inherent mechanism within the cell for effectively controlling these initial reactions, thereby regulating the entire metabolic network efficiently. Since these initial reactions primarily involve flux-generating nonequilibrium reactions (Court et al, 2015; Dai and Locasale, 2018; de Hijas-Liste et al, 2015), they are frequently regulated to facilitate pathway flux (Microbiology, 2015). The intracellular activation of key enzymatic reactions in most biochemical pathways is a response to the imperative requirement for precise regulation and control of cellular metabolic processes (Lopina, 2017; Metallo and Vander Heiden, 2013; Millard et al, 2017; Zhu and Thompson, 2019). Interestingly, we also observe activated enzymes at various positions in all biochemical pathways, even though the activation of individual enzymatic reactions might not consistently benefit the entire pathway (Fig. 2Bi). One rationale for this finding is that around 27% of enzymatic reactions found in the middle or later stages of a pathway are also commonly present at the start of other biochemical pathways (Fig. 2Bi). Activating the remaining non-initial enzymatic reactions could be essential for maintaining flux through the pathway after the initial reactions are activated (de Hijas-Liste et al, 2015; Robinson, 2015). Moreover, the presence of activated enzymes at different positions in biochemical pathways underscores the complex and adaptable nature of cellular metabolism (de Hijas-Liste et al, 2015).

This configuration empowers cells to meticulously regulate their metabolic activities, respond to diverse signals, and optimize their functionality in various scenarios (Chubukov et al, 2012; de Hijas-Liste et al, 2015; Riehl and Segrè, 2008).

We further explored the extent to which metabolites from one pathway are significantly enriched to activate enzymes in other pathways (Fig. 2C). This investigation is prompted by the prevailing allosteric binding of activators with enzymes, facilitating the binding of substrates to the enzyme (Tzeng and Kalodimos, 2009). It is noteworthy that activator metabolites and substrates exhibit structural dissimilarity owing to this widespread allosteric binding mechanism. In addition, prior research has illustrated that metabolites within the same pathways demonstrate greater structural similarity in comparison to those across different pathways, often leading to their association with competitive binding with enzymes—the most common inhibitory interactions (Alam et al, 2017). Interestingly, the configuration of the activation interactions network between pathways stands in stark contrast to that of the human inhibitory interactions network. In the latter, products originating from the same pathway often act as significant inhibitors (Alam et al, 2017), whereas the activatory interactions between pathways are all cross-pathway in nature (Fig. 2C). This disparity underscores the distinct nature of activation and inhibition interactions within the metabolic network, playing a crucial role in constraining metabolism for optimal efficiency.

In terms of activation of different enzyme classes, transferases, oxidoreductases, and hydrolases, which are also the most prevalent enzyme classes in the metabolic network in that order, cover most of the activated enzymes. In particular, hydrolases and oxidoreductases have a higher prevalence of activated enzymes compared to non-activated enzymes (Fig. 1F), which could be due to the nature of their catalytic activities and the functional requirements of these enzyme classes. Hydrolases, for instance, catalyze the hydrolysis of various substrates by adding water molecules (Morisseau, 2022). Activators can modulate the enzyme's conformation, making it more receptive to substrate binding and efficient hydrolysis. Similarly, oxidoreductases are involved in oxidation-reduction reactions, fundamental to cellular processes, where electrons are transferred between substrates (May, 1999). Activation of these enzymes by specific metabolites allows for precise control of electron transfer and redox balance in response to cellular needs. In contrast, isomerases and ligases, which are the least prevalent enzymes in the metabolic network, have an even lower percentage of activated enzymes (Fig. 1F). These enzyme classes typically engage in enzymatic activities that are inherently less complex or less dependent on allosteric regulation. Isomerases facilitate structural rearrangements, which may not require extensive external modulation. Ligases, on the other hand, often have an inherent regulatory mechanism through the energy input required for bond formation (McDonald and Tipton, 2023). This over or under-representation of intracellular activation highlights the importance of activation for specific classes of enzymatic reactions.

Further, within the list of activated enzymes, we find 30% of enzymes are essential for growth in a minimal media condition; a similar proportion of essentiality was also observed for non-activated enzymes (34%) (Fig. 3A). Strikingly, the essential activated enzymes have substantially less degree of activation compared to non-essential enzymes (Fig. 3C). This includes the topmost activated enzymes (>20 activation interactions) such as pyruvate kinase (2.7.1.40), glutamate dehydrogenase (1.4.1.2),

**Table 1.  Top 10 intracellular activating metabolites and activated enzymes.**

| Top 10 cellular-activating metabolites | | | | Top 10 cellular-activated enzymes | | | |
|---|---|---|---|---|---|---|---|
| Activators ID | Activators name | # activation interactions | Essentiality | EC | Enzyme name | # activation interactions | Essentiality |
| C00097 | L-cysteine | 67 | Y | EC:1.4.1.2 | Glutamate dehydrogenase | 28 | N |
| C00002 | ATP | 61 | Y | EC:2.7.1.40 | Pyruvate kinase | 28 | Y |
| C00051 | Glutathione | 59 | N | EC:3.5.4.1 | Cytosine deaminase | 25 | N |
| C00009 | Orthophosphate | 43 | Y | EC:3.1.3.11 | Fructose-bisphosphatase | 24 | N |
| C00469 | Ethanol | 36 | N | EC:3.1.3.2 | Glycerophosphatase | 23 | N |
| C00157 | Phosphatidylcholine | 27 | Y | EC:3.1.4.4 | Phospholipase D | 21 | N |
| C00008 | ADP | 27 | Y | EC:3.1.2.2 | β-D-glucosides | 19 | Y |
| C00020 | AMP | 24 | Y | EC:3.1.2.1 | Acetyl-CoA hydrolase | 18 | N |
| C00750 | Spermine | 24 | Y | EC:2.4.1.11 | Glycogen synthase | 17 | Y |
| C00350 | Phosphatidylethanolamine | 24 | Y | EC:2.7.1.11 | 6-phosphofructokinase | 17 | N |

cytosine deaminase (3.5.4.1), fructose-bisphosphatase (3.1.3.11), glycerophosphatase (3.1.3.2), and phospholipase D (3.1.4.4) which are intracellularly activated by 28, 28, 25, 24, 23, and 21 different cellular metabolites, respectively, where 5/6 are non-essential for growth. This remarkable significant difference in the degree of activation between essential and non-essential enzymes may be a result of the evolutionary and functional characteristics of these enzymes (Chubukov et al, 2012; de Hijas-Liste et al, 2015). The role and involvement of essential enzymes in core cellular processes might necessitate a more streamlined and efficient mode of regulation, with less reliance on activator interactions.

We then examined which cellular metabolites are optimal to act as an activator. For cellular activator metabolites, we find strikingly opposite features compared to activated enzymes (Fig. 3). We observe that the ratio of essential metabolites among the activators is significantly higher compared to non-activator metabolites (45% essential activators vs 11.2% essential non-activators) (Fig. 3A). In addition, the essential metabolites activate a significantly higher number of enzymatic reactions compared to non-essential metabolites (Figs. 3B and  1E). Some of the highly activating metabolites are L-cysteine (C00097), ATP (C00002), glutathione (C00051), orthophosphate (C00009), ethanol (C00469), phosphatidylcholine (C00157), ADP (C00008), AMP (C00020), spermine (C00750), phosphatidylethanolamine (C00350), glycerol (C00116), activating >20 enzymes, where 8/11 activators are essential for growth (Table 1). Since essential metabolites are consistently generated within the cell, regardless of the conditions, employing them to also activate other enzymes might be a strategic and resource-efficient approach (Wessely et al, 2011). Furthermore, the conservation of crucial metabolic pathways could also contribute to the predominance of essential metabolites serving as activators. Moreover, in nearly one-third of the activatory interactions, a gene-producing activator is co-expressed with activated enzymes. This may be associated with positive feedback loops or co-regulated gene networks, where the activator enhances the expression of enzymes involved in a biochemical pathway. For example, transcriptional activators can be co-expressed with the enzymes they regulate. Similarly, in certain signaling cascades, an activator protein may regulate its own expression. In addition, we observe that in one-third of the activatory interactions, a gene-producing activator also engages in genetic interactions.

Lastly, we computed the shortest path length for all cellular metabolites from different nutrient sources, including glucose. Our findings indicate that activators exhibit significantly shorter path lengths to nutrients compared to non-activators (Fig. 4). This suggests that activator metabolites are promptly produced following nutrient uptake to regulate enzymes distributed throughout the entire metabolic network. The early production of activator metabolites after nutrient uptake underscores a coordinated and energy-efficient strategy employed by cells to modulate metabolic pathways. This timely response ensures a rapid and effective adaptation to nutrient availability, enabling cells to generate metabolic flux and efficiently utilize nutrients for diverse cellular processes.

## Methods

**Reagents and tools table**

| Reagent/resource | Reference or source | Identifier or catalog number |
|---|---|---|
| **Experimental models** | | |
| **Recombinant DNA** | | |
| **Antibodies** | | |
| **Oligonucleotides and other sequence-based reagents** | | |
| **Computational model** | | |
| Yeast9 | https://github.com/ SysBioChalmers/yeast-GEM | Yeast9 |
| **Software** | | |
| igraph package in R | https://cran.r-project.org/ web/packages/igraph/ index.html | |
| COBRA Toolbox v.3.0. | https://opencobra.github.io/ cobratoolbox/stable/ index.html | |
| limma package in R | https://bioconductor.org/ packages/limma | |

| Reagent/resource | Reference or source | Identifier or catalog number |
|---|---|---|
| BRaunschweig ENzyme DAtabase (BRENDA) | https://www.brenda-enzymes.org | |
| The Chemical Translation Service (CTS) | http://cts.fiehnlab.ucdavis.edu | |
| KEGG database | https://www.genome.jp/brite/sce00001.keg | |
| Other | | |

## Reconstructing the cell-intrinsic activation network of *Saccharomyces cerevisiae*

The cell-intrinsic enzyme–metabolite activation-interaction network ('activation network') of *Saccharomyces cerevisiae* was reconstructed by combining the activating compounds, collected from the BRaunschweig ENzyme DAtabase (BRENDA) database (Data ref: Chang et al, 2021), with the *S. cerevisiae* genome-scale metabolic model (Data ref: Zhang et al, 2023). The list of all possible activating compounds for every metabolic enzyme of the *S. cerevisiae* metabolic model (total of 635 unique EC numbers) was downloaded from the BRENDA database using the *getActivatingCompound* function following SOAP clients instructions (Data ref: Chang et al, 2021). In total, 2645 compounds, which included both cellular and non-cellular compounds, were reported to activate 465 metabolic enzymes. In addition, several activating compounds appear to use multiple synonyms in the BRENDA database, and this redundancy was removed after curation. A unique ID from the Kyoto Encyclopedia of Genes and Genomes (KEGG) database (Kanehisa et al, 2016) was assigned to each activating compound and Yeast9 model compound using a combination of the chemical translation service (http://cts.fiehnlab.ucdavis.edu/) and manually examining the KEGG database synonyms (Kanehisa et al, 2016). Then, all non-cellular compounds were removed from the analysis by mapping the total activating compounds with the list of metabolic compounds of the model. In total, 286 metabolites were successfully mapped to Yeasy9 metabolites, activating 344 enzymes. Finally, the activation network was created by joining enzymes with metabolite nodes when an enzyme is reported to be activated by a metabolite, constituting a total of 1499 edges in the network (Dataset EV1). Based on the frequency of activatory interactions, that is the number of edges for each node, the degree of activation for enzyme or activator metabolite nodes was calculated. For example, if an enzyme is activated by $n$ metabolites, the degree of activation for that enzyme is $n$. Similarly, if a metabolite activates m different enzymes, the degree of activation for that metabolite is m.

## Enzyme and metabolite essentiality prediction

The essentiality of enzymes and metabolites was predicted by performing in-silico knockout experiments on the Yeast9 model using Cobra Toolbox (Heirendt et al, 2019). For an enzyme (or metabolite) knockout, both the lower and upper flux bounds of all associated reactions were set to zero. Then, the maximum growth was predicted by the Flux Balance Analysis (FBA) approach on a minimal media constraint setting. If the predicted growth in the

enzyme (or metabolite) knock-out condition was reduced by more than 90% compared to the normal condition growth rate, then the enzyme (or metabolite) was considered essential. The default minimal media setting of the model was used for simulation.

## Enzyme and metabolite classes

For enzyme and metabolite classification, the KEGG database definition was used (Kanehisa et al, 2016). The enzymes were grouped into 6 enzyme classes including oxidoreductases, transferases, hydrolases, lyases, isomerases, ligases, and translocases. For metabolite classification, we used the KEGG database definition. Various small compound groups were merged into one larger group. In total, we use 8 metabolite group nomenclature.

## Biochemical pathway activation

Biochemical pathway information was taken from the KEGG database (Kanehisa et al, 2016). For every pathway, the order of enzymatic reactions was based on KEGG Orthology (KO)—*Saccharomyces cerevisiae* (https://www.genome.jp/brite/sce00001.keg). The enzymes which are not present in the metabolic model reconstruction were removed.

## Network of pathway–pathway activation interactions

The pathway–pathway activation network was created based on the biochemical pathway definitions in the Yeast9 metabolic model (Data ref: Zhang et al, 2023). First, for each pair of pathways, the activation frequency was calculated by counting the number of metabolites produced in one pathway that activated the enzymes of another pathway, excluding metabolites common to both pathways. This activation frequency was calculated for all pairs of metabolic pathways, resulting in a comprehensive table. Then, using a hypergeometric statistical test, the significance of activation between each pair of pathways was determined. Finally, the pathway–pathway activation network was reconstructed by linking two pathways if the $P$ value for one pathway activating another was less than 0.05.

## Transcriptome data analysis and gene co-expression

The raw microarray gene expression data from 207 different studies from the Genome2 chips, covering 3115 samples and 665 unique experimental conditions, were extracted from the ArrayExpress database (Data ref: Parkinson et al, 2007) and processed by applying the limma package functions in R (Ritchie et al, 2015). The RAW cel files were processed, and the average normalized gene expression data were calculated for each unique condition. Using the transcriptome profile, the Pearson correlation test was performed for each pair of genes by the *rcorr* function in R. Adjusted $P$ value was calculated by Benjamini–Hochberg (BH) multiple testing procedure using the *P.adjust* function. Two genes were considered co-expressed if $r > 0.75$ and adjusted $P$ value < 0.001.

## Shortest path length for metabolites and enzymes in the metabolic network

In a graph, the shortest path length is the minimum number of edges between the two nodes. In order to calculate the shortest path

length for metabolites, first, the genome-scale model (Zhang et al, 2023) was converted into a graph where nodes represent the metabolite and edges are formed when metabolites are connected by a chemical reaction. Then, highly connected metabolites, having more than 50 metabolic interactions in any single cellular compartment, were removed (list of removed metabolites are in Dataset EV1). Then, for every metabolite, the shortest path length from glucose metabolite was calculated by counting the minimum number of edges between the two metabolites. Similarly, to calculate the shortest path length for enzymes, the Yeast9 model (Zhang et al, 2023) was converted into a graph where nodes represent the reactions/enzymes, and edges are established between nodes if a substrate of one reaction is a product of another reaction/enzyme. For each reaction/enzyme, the minimum number of edges from the glucose transport reaction was considered the shortest path length of the reaction/enzyme. The shortest path length for metabolites and enzymes were calculated by *shortest.paths* function from the igraph package in R.

### Genetic interactions comparison

The genetic interaction data were obtained from Costanzo et al (Costanzo et al, 2010).

### Computational analysis

Networks were created using functions from the igraph package in R. All the statistical analyses were performed in R. The cobra toolbox was used for simulating genome-scale metabolic reconstruction (Heirendt et al, 2019). Gene expression data were processed using the limma package in R (Ritchie et al, 2015).

## Data availability

This study includes no new experimental data deposited in external repositories. The datasets and computer code produced in this study are available in the following repository (GitHub): http://github.com/mdtauqeer/activationNetwork.

The source data of this paper are collected in the following database record: biostudies:S-SCDT-10_1038-S44320-025-00111-7.

## Peer review information

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

## Acknowledgements

SM Al-Zubaidi, MIN, and MTA were supported by the UAE University internal research grants (startup grant code G00003688, UPAR grant code G00004152, and Strategic Research Program G00004540).

## Author contributions

**Sultana Mohammed Al Zubaidi**: Data curation; Formal analysis; Investigation; Methodology. **Muhammad Ibtisam Nasar**: Data curation; Software; Formal analysis; Investigation; Methodology. **Richard A Notebaart**: Investigation; Methodology; Writing—review and editing. **Markus Ralser**: Investigation; Methodology; Writing—review and editing. **Mohammad Tauqeer Alam**: Conceptualization; Data curation; Software; Formal analysis; Supervision; Funding acquisition; Investigation; Visualization; Methodology; Writing—original draft; Project administration; Writing—review and editing.

Source data underlying figure panels in this paper may have individual authorship assigned. Where available, figure panel/source data authorship is listed in the following database record: biostudies:S-SCDT-10_1038-S44320-025-00111-7.

## Disclosure and competing interests statement

The authors declare no competing interests.

# Expanded View Figures

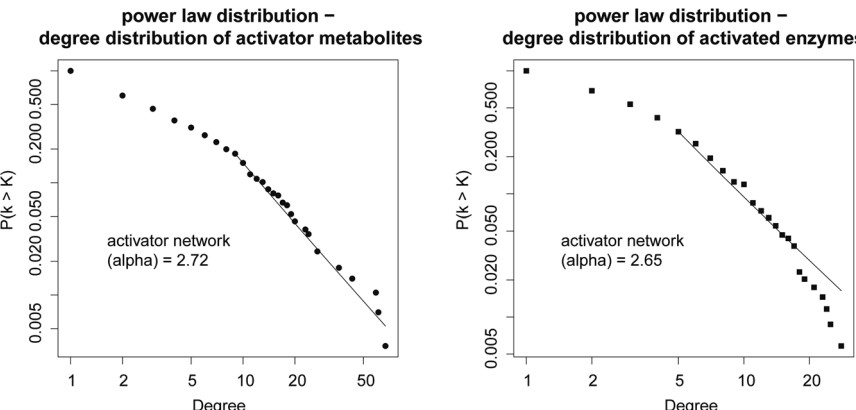

**Figure EV1.    Degree distribution of nodes in the activation network.**

Degree distribution of activator metabolites and activated enzymes in the cell-intrinsic activation interaction network follow power law distribution.

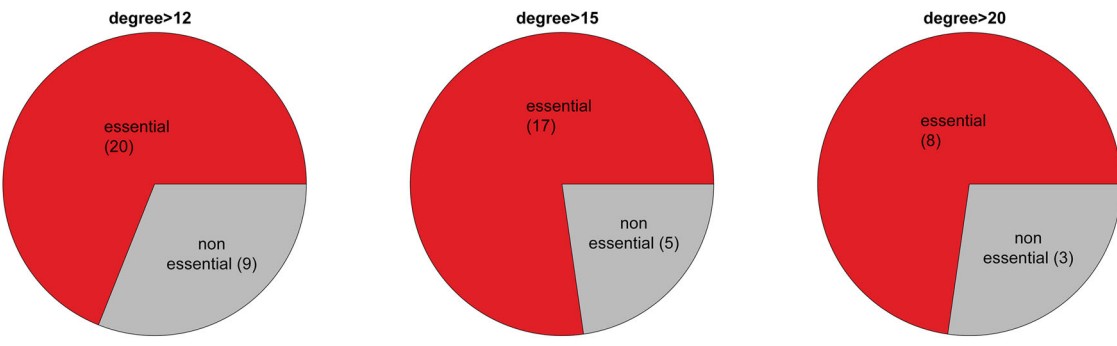

**A** Degree of activation for highly activating hub metabolites

degree>12

essential (20)

non essential (9)

degree>15

essential (17)

non essential (5)

degree>20

essential (8)

non essential (3)

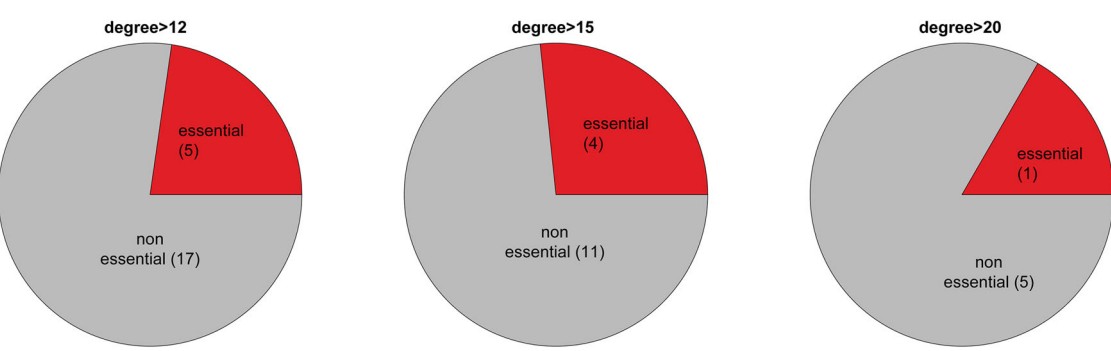

**B** Degree of activation for highly activated hub enzymes

degree>12

essential (5)

non essential (17)

degree>15

essential (4)

non essential (11)

degree>20

essential (1)

non essential (5)

**Figure EV2. Essentiality of highly connected activators and enzymes.**

(A) The degree distribution of activator metabolites within the cell-intrinsic activation interaction network reveals that highly interactive activators (degree > 12, degree > 15, degree > 20) are mostly essential for growth. (B) The degree distribution of activated enzymes within the cell-intrinsic activation interaction network reveals that highly activated enzymes (degree > 12, degree > 15, degree > 20) are mostly non-essential for growth.

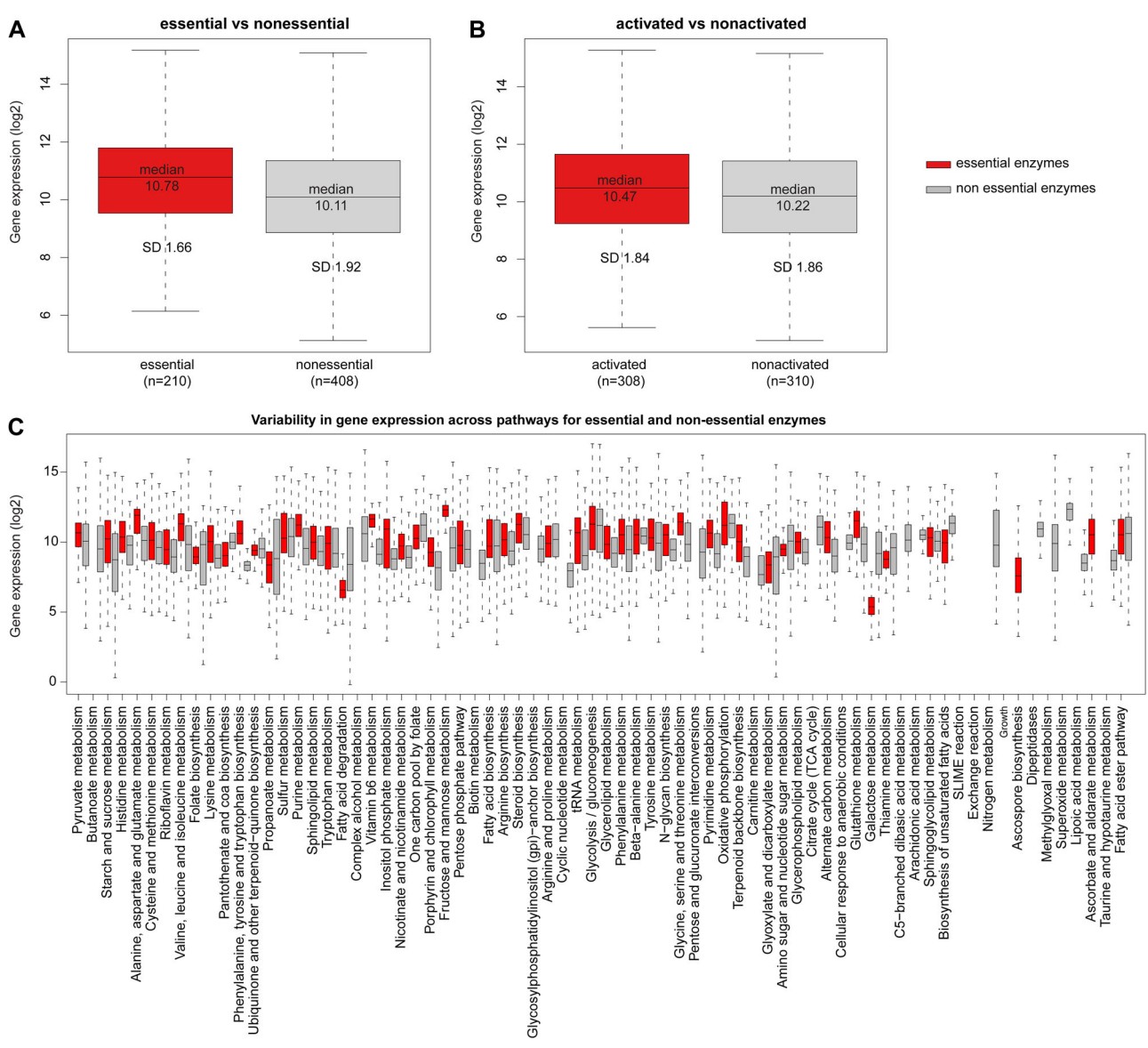

**Figure EV3. Variation in gene expression across different enzyme groups.**

Gene expression variation among (**A**) essential vs non-essential enzymes, and (**B**) activated vs non-activated enzymes. (**C**) Pathway wise gene expression variation between essential and non-essential enzymes: For each metabolic pathway, a set of essential and non-essential enzymes were identified using in silico enzyme knockout experiments and gene expression variation between essential and non-essential enzymes was calculated using the public data obtained from microarray gene expression experiments. Box plots display the median, the upper and lower quartiles, and the minimum and maximum values represented by the whiskers.

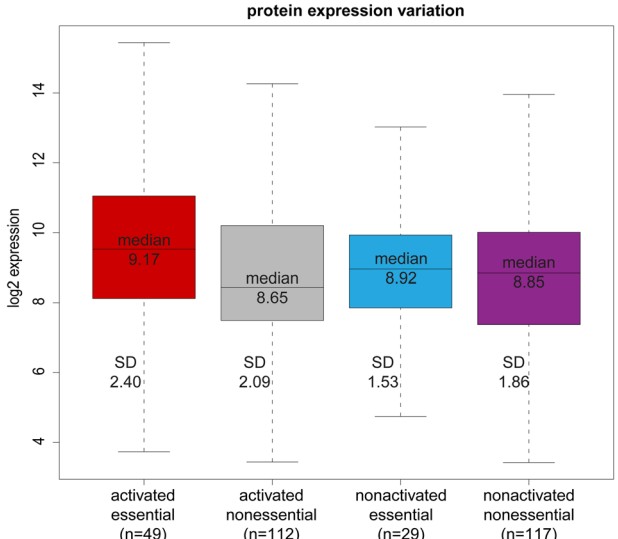

**Figure EV4.   The protein expression variation among activated essential enzymes, activated non-essential enzymes, non-activated essential enzymes, and non-activated non-essential enzymes.**

Box plots display the median, the upper and lower quartiles, and the minimum and maximum values represented by the whiskers.

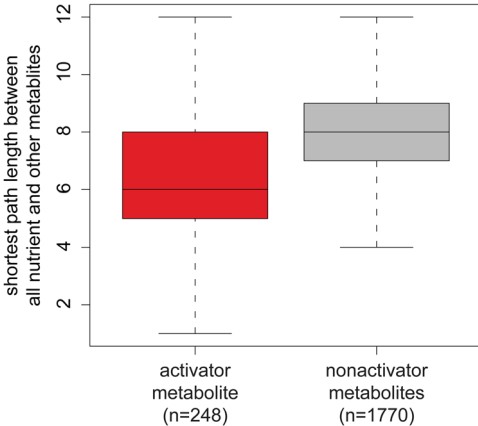

**metabolite shortest path length in the enzyme-metabolite activation network**

**Figure EV5.  Shortest path length of activator and non-activator metabolites within the cell-intrinsic activation interactions network from all nutrient compounds.**

Box plots display the median, the upper and lower quartiles, and the minimum and maximum values represented by the whiskers.

