## [Peer Review File · Molecular Systems Biology]

An enzyme activation network reveals extensive regulatory crosstalk between metabolic pathways

Sultana Al Zubaidi, Muhammad Ibtisam Nasar, Richard Notebaart, Markus Ralser, and Mohammad Tauqeer Alam

Corresponding author(s): Mohammad Tauqeer Alam (mtalam@uaeu.ac.ae)

Review Timeline:

Transfer from Review Commons:	14th Jan 25
Editorial Decision:	26th Feb 25
Revision Received:	16th Mar 25
Editorial Decision:	10th Apr 25
Revision Received:	14th Apr 25
Accepted:	24th Apr 25

Editor: Poonam Bheda

Transaction Report:

Review #1

1. Evidence, reproducibility and clarity:

Evidence, reproducibility and clarity (Required)

The authors investigate intracellular allosteric interactions between metabolites and enzymes in *Saccharomyces cerevisiae*. Briefly, the methodology consists of utilizing graph theory to construct links between metabolites and enzymes and to color them with various classifications such as activator, non-competitive inhibitor, essential or non-essential. Against this network was performed tests to examine various metrics and statistical significance. The notable results from the analyses indicate that: (i) enzymes that have activation interactions are mostly found within the first three enzymes of any pathways, (ii) a majority of the central metabolic pathways are activating or activate other pathways, (iii) metabolites that are essential for growth are mostly activators in the network mostly activate nonessential enzymes, and (iv) activator metabolites are produced in the early steps (4-5) of any given pathway.

Although the area of investigation is important and of interest, the work in its current form falls short of adequately addressing the problem at hand and has a number of flaws. Overall, this manuscript lacks finer details with specific and persuasive arguments. For instance, the authors often state data or results for a process but fail to deliver any further information or hypothesis about why/how and their implications.

p4. What was the rationale used to select the *Saccharomyces cerevisiae* genome-scale metabolic network? IMM904 was published in 2009 and there have been continued updates to the consensus genome-scale metabolic model of *S. cerevisiae*, Yeast GEM, with Yeast8 being released in 2019. Notably, Yeast9 was made available in 2023 and its bioRxiv preprint article doi.org/10.1101/2023.12.03.569754 involves integrative multi-omics analysis. Although the methodology and framework do not necessarily depend on the network, the authors should clearly address the design choice in model selection and what features might be missed that are covered in other models such as Yeast8.

p4. The authors cite a work involving human enzyme inhibition interactions but then make a comparison with activation interactions. It is not clear how the citation (or inclusion of human enzymes) is relevant.

p5. What are the criteria for the degree of activation in Figure 2b? The authors should make their selection clear, including explicitly delineating and justifying them.

p6. The authors state that "most of the central metabolic pathways are significantly activating or activated by other pathways". The pathway analysis (as described in the Methodology section) simply counts the number of metabolites produced from one pathway that activates the enzymes of another pathway. Are the set of all metabolites in each pathway unique or do they overlap as substrates/products in other pathways as well? Related to this question is how do the authors take into account the fact that metabolites can occur in more than one pathway?

p6. The author finds "In concordance with other studies, we find that the essential enzymes have..." but does not cite any studies. Please cite them. The author then states that essential enzymes have higher gene expression than non-essential enzyme Can the comparison be evaluated by comparing essential vs. non-essential enzymes on a pathway-to-pathway basis?

p6. The author state that "NAD biosynthesis is activated by two different pathways" but does not provide further insight on how this happens. In general, this lack of insight , and instead a rote listing of results, (both here and elsewhere) is a key detriment to the impact of the manuscript.

p7. Standard deviations are not reported for all averages of gene expression. What is the measurement error on the average gene expression levels and are the differences significant? If the standard deviations that are not reported are similar to those that are, then the differences cannot be considered significant.

p7. In the results reported on the previous page the author comes to the conclusion by comparing the average gene expression. However, even if one ignores the statistical matter, it is not clear how gene expression level can be used here, as differences in translation can result in different protein levels. The authors should clearly justify their use of expression data and raise all the limitations that brings. Are there protein level data that could better support the hypotheses? The manuscript would be substantially improved by their incorporation.

p7. Although the authors do remove highly connected co-factors such as ATP or NAD/NADPH, etc. from the graph with nodes representing metabolites connected by chemical reactions contain, but that fact is buried in the methods and should be mentioned. Also, in the methods the phrase "Some of the highly connected metabolites including..." occurs but for reproducibility all of the excluded metabolites should be

explicitly listed. How did the authors define a "highly connected metabolite"?

p8. Once again, nebulous "other studies" are mentioned but not cited, and the list is reported as "well-characterized" without defining what that means. One of the studies the authors refer to is probably Noor et al. 2010 (10.1016/j.molcel.2010.08.031) but please actually cite all the studies referred to. Moreover, the identity of these 12 metabolites and how they differ (presumably, since the authors state "similar to this overservation") and the implications thereof.

p8. The authors use graph theory to look at the shortest path for activators and simply state that "majority of the activator metabolites have the shortest path length of 5 whereas for non-activator this distance ranges from 5-7". How many are these exactly out of how many (that is, give the result as a non-reduced fraction or ratio). Is this shortest path accounting for reactions that only produces intermediate metabolites not used elsewhere in the system (e.g. aconitase) or the presence of multienzyme complexes that are modeled with several reactions (e.g., fatty acid synthase)?

p8. The authors very briefly bring in an analysis of microarray gene expression data from the ArrayExpress database. However, no mention of controlling for transcription level regulation is mentioned in the methods. How are the authors confident that what is observed can be attributed solely to allosteric regulation? The case for their conclusions is not sufficiently made and the authors should reevaluate the results after taking into account transcription regulation. Moreover, protein levels do not exactly correlate with gene expression data, and the authors do not bring up this caveat.

p9. The authors assume as ground truth that all interactions given in BRENDA are present in their network. The authors do correctly note in the discussion that they might (daresay are) overestimating the number of activators. Was any attempt made to examine yeast (or even better, *S. cerevisiae*) studies to validate or curate the information from BRENDA? Were proteins across all clades of life used? Another avenue could be to use protein structures (predicted or measured) to evaluate if there are binding sites for the metabolites.

p12. The author states in that activator information is directly pulled from BRENDA using a method akin to a web scraping tool (SOAP). The BRENDA database often does not accurately provide sufficient information for curating allosteric information as they often depend on growth conditions. Did the authors check with each activator used in the network model and verify the activator metabolite for the given experimental condition? For the 655 unique datasets examined, were filters added to distinguish aerobic vs anaerobic

conditions? The authors should clearly comment on in the manuscript on how much of their network and results are condition dependent.

****Minor points:****

p1. Affiliation 5 is labeled with the number 2 in the author list.

p2. The authors should consider turning on line numbers.

p4. The method for downloading BRENDA (i.e., SOAP) is listed in the Results section but not mentioned in Methodologies. Since it is a method, this detail should be moved to Methodologies.

p6. The authors states "several central metabolic pathways (15/18) are..." but does not define which 18 pathways are classified as the central metabolic pathways. Please define them in the methods section or include them in supplemental information.

p6. The authors use the term 'hub activator' and the wording suggests that they have defined it as a node that has activation connections with more than 15 enzymes but the sentence should be rephrased to make that clear. Also, how was the cutoff of 15 selected? Do the results change if a number such as 12 or 20 or top 10% is used?

p7. How does the non-activator distance range from 5-7 if the reported maximum length is 6?

p8. The authors state "Figure 4a shows a subnetwork of the main cell-intrinsic activation interaction" however the reference should probably be Figure 5a since Figure 4a does not show a subnetwork but Figure 5a does.

p9. Human metabolism is again mentioned yet its relevance is unclear.

2. Significance:

Significance (Required)

The manuscript would require significant changes before acceptable for publication.

3. How much time do you estimate the authors will need to complete the suggested revisions:

Estimated time to Complete Revisions (Required)

(Decision Recommendation)

Cannot tell / Not applicable

4. Review Commons values the work of reviewers and encourages them to get credit for their work. Select 'Yes' below to register your reviewing activity at Web of Science Reviewer Recognition Service (formerly Publons); note that the content of your review will not be visible on Web of Science.

No

Review #2

1. Evidence, reproducibility and clarity:

Evidence, reproducibility and clarity (Required)

The authors generate and analyse a genome-scale metabolic network of allosteric activation in *S. cerevisiae*. This is a very interesting and relevant piece of work that sheds light upon the role of allosteric interactions, in particular allosteric activation, in the regulation of metabolism.

****Major issues:****

- One major problem with allosteric regulation is that we still do not have an unbiased high-throughput method to completely cover the whole enzyme-metabolite combinatorial space. Entries in databases like BRENDA correspond to in vitro studies that are biased towards well-studied pathways. Therefore, it is not surprising that core pathways like central carbon, nucleotide, and amino acid metabolism seem to be the mostly regulated. The authors should at least discuss this issue.
- Additionally, the authors could try to mitigate this issue by normalizing the number of reported allosteric interactions for a metabolite or an enzyme by either the corresponding number of literature references in BRENDA for the same metabolite/enzyme or by the number of reported inhibitory interactions. Are there metabolites that are frequent activators but not frequent inhibitors? Or reactions that are frequently activated but not

frequently inhibited?

- Enzyme essentiality was computed on glucose minimal medium, which is an artificial laboratory setting and not a realistic scenario for species living in nature. Therefore, there is no distinction between conditionally essential enzymes and the universally essential enzymes (those that would also be essential in a complete medium). I think it would be important to make this distinction and test if it has any effect on the results presented.

- Also, I wonder why authors did not simply use in vivo essentiality data (which can be found in the literature for such a well-studied organism like this one) instead of predicted in silico data.

- In the first paragraph of page 7 the authors describe the differences in gene expression observed in figure 3c(iii). I would argue that in fact the conclusion is that there are no observable differences. It is important to note that magnitude of difference and statistical significance are different things. These differences might be statistically significant (due to large sample size), but they are of very small magnitude. One can conclude, with statistical confidence, that there is almost no difference.

- I am a bit skeptic of the results regarding pathway length. How is it possible that most metabolites are on average only 5 steps away from glucose, and most enzymes on average only 4 steps away? I would expect most metabolites and enzymes to be tens of steps away. For instance, the aromatic amino acid pathway is several steps long, how would tryptophan be only 5 steps away from glucose? I suspect there are several shortcuts due to the presence of co-factors in the graph generating algorithm. The authors say they remove "some of the highly connected metabolites like h, h₂o, adp, atp, amp, pi, nad, nadh". This part needs a more rigorous description. All cofactors (including redox pairs, CoA groups, etc) should be removed.

- The authors mention that they used all allosteric activations in BRENDA, not only those reported for *S. cerevisiae*, assuming that allosteric regulation is conserved. In fact, there are some articles arguing that allosteric regulation might not be under strong selective pressure, and therefore might not be as conserved as we might think. I would strongly recommend building a network using only interactions reported at the same species (or genus, or any relatively conserved taxonomic) level, and re-running some of the tests just to confirm that they don't affect the main conclusions.

- It would be interesting to include in the discussion some comments about why evolution might favor allosteric activation instead of transcriptional regulation to increase enzyme activity. On one hand, allosteric activation is faster, and would allow the cell to respond more rapidly to environmental changes. On the other hand, producing enzymes which are unused and waiting for activation is more costly. There are many recent papers showing how protein production cost is the main metabolic constraint limiting growth. It would be interesting to compare the degree of enzyme regulation with the respective protein cost.

- In the discussion section, please replace all the EC numbers and KEGG ids listed with enzyme and metabolite names. In fact, it would be nice to have a table in the main text with the top 10 activator metabolites and top 10 activated enzymes.
- I am curious why the authors used the outdated iMM904 model (Mo et al, 2009) and not a more recent model like Yeast8 (Lu et al, 2019).
- Please provide a repository link with all the code and data so that readers can try to reproduce your results.

****Referees cross-commenting****

It seems that there are some points in which all reviewers are in agreement, especially when it comes to improve the methodology description, such as the download of non-yeast data from BRENDA, the use of an outdated yeast model, and a more clear definition of pathways and pathway distance.

Nonetheless, I think it is quite a nice and relevant piece of work. I look forward to see it published.

2. Significance:

Significance (Required)

I find this study to be of high significance. It explores an often overlooked aspect of metabolic regulation.

3. How much time do you estimate the authors will need to complete the suggested revisions:

Estimated time to Complete Revisions (Required)

(Decision Recommendation)

Between 1 and 3 months

Yes

Review #3

1. Evidence, reproducibility and clarity:

Evidence, reproducibility and clarity (Required)

Zubaida et al mapped known information of metabolic activators onto a genome scale metabolic model to address the importance of activation and its locality. I found the study to be excellent and clear. I do not feel that reproducibility is a criteria that can easily be addressed in such studies.

I only have three minor issues which I address in order of importance.

1. the authors should clarify if the activator information they use from BRENDA is for yeast specifically or merely in ANY organism. Similarly was it collected in physiologically relevant conditions or at the enzymes optimal conditions. I feel this information should be provided as they may effect the outcome somewhat. The latter cannot be helped as it was how enzymology typically was done but this should be discussed.
2. the authors talk about activation by extracellular molecules I assume that they are implying that these are physiologically irrelevant however this should perhaps be explicitly stated.
3. at many times metabolites start with an uppercase letter this is only correct when they are also the first word of a sentence.

2. Significance:

Significance (Required)

This is a highly important study with broad implications regarding the regulation of metabolism. I found it clearly worded, well planned and well executed.

3. How much time do you estimate the authors will need to complete the suggested revisions:

Estimated time to Complete Revisions (Required)

(Decision Recommendation)

Less than 1 month

Yes

Full Revision

Manuscript number: RC-2023-02262

Corresponding author(s): Mohammad Tauqeer Alam

[Please use this template only if the submitted manuscript should be considered by the affiliate journal as a full revision in response to the points raised by the reviewers.]

*If you wish to submit a preliminary revision with a revision plan, please use our "Revision Plan" template. **It is important to use the appropriate template to clearly inform the editors of your intentions.**]*

1. General Statements [optional]

We sincerely thank all the reviewers for their positive assessment of our manuscript and for their valuable critical suggestions. In particular, they noted our use of an older model in the initial submission, and we fully agree with their observations. In response, we have revised the study using the latest version of the metabolic network as the foundation for this paper. As a result, we have updated all figures and numerical values. Importantly, the core message of the study remains unchanged. We believe the manuscript has been greatly enhanced through the reviewers' input. We have conducted new analyses to address all of their comments and have rewritten several sections to improve clarity.

Reviewer #1 (Evidence, reproducibility and clarity (Required)):

The authors investigate intracellular allosteric interactions between metabolites and enzymes in *Saccharomyces cerevisiae*. Briefly, the methodology consists of utilizing graph theory to construct links between metabolites and enzymes and to color them with various classifications such as activator, non-competitive inhibitor, essential or non-essential. Against this network was performed tests to examine various metrics and statistical significance. The notable results from the analyses indicate that: (i) enzymes that have activation interactions are mostly found within the first three enzymes of any pathways, (ii) a majority of the central metabolic pathways are activating or activate other pathways, (iii) metabolites that are essential for growth are mostly activators in the network mostly activate nonessential enzymes, and (iv) activator metabolites are produced in the early steps (4-5) of any given pathway.

Although the area of investigation is important and of interest, the work in its current form falls short of adequately addressing the problem at hand and has a number of flaws. Overall, this manuscript lacks finer details with specific and persuasive arguments. For instance, the authors often state data or results for a process but fail to deliver any further information or hypothesis about why/how and their implications.

Response: We would like to thank the reviewer for their detailed comments and suggestions and for accurately summarizing the manuscript. We have taken their feedback and suggestions seriously and addressed them through clarifications, revisions to the main text, and additional analysis. We hope to have been able to address all points the reviewer made. We have put particular emphasis in the revision, to put more hypotheses forward about what the key observations can mean. Our detailed point-by-point response is provided below.

p4. What was the rationale used to select the *Saccharomyces cerevisiae* genome-scale metabolic network? iMM904 was published in 2009 and there have been continued updates to the consensus genome-scale metabolic model of *S. cerevisiae*, Yeast GEM, with Yeast8 being released in 2019. Notably, Yeast9 was made available in 2023 and its bioRxiv preprint article doi.org/10.1101/2023.12.03.569754 involves integrative multi-omics analysis. Although the methodology and framework do not necessarily depend on the network, the authors should clearly address the design choice in model selection and what features might be missed that are covered in other models such as Yeast8.

Response: We would like to thank the reviewer for highlighting this important issue. Other reviewers have also commented on this matter, so we have decided to redo our study using the more recent Yeast9 model. As a result of the re-analysis, all results in the main manuscript have changed, and we have now revised all relevant Figures and text accordingly. However, we report that the conclusions remained essentially unchanged; the small changes in topology between iMM904 and the newer model had no impact on the main findings and conclusions of our study.

p4. The authors cite a work involving human enzyme inhibition interactions but then make a comparison with activation interactions. It is not clear how the citation (or inclusion of human enzymes) is relevant.

Response: We understand the reviewer's point but would however like to highlight that in both cases, we deal with cross species networks - only the topology differs in one case on the human and in the other case on the yeast network. This strategy was necessary (in both studies), to achieve a significant coverage of the enzymatic reactions; by focusing on just one species alone, one does not reach a significant data coverage. It follows that the assumption that we and others have thus taken in this and previous work, is to assume that many metabolic reactions and their properties are conserved across species. We are aware that this is not perfect, and thus a known limitation of our study (see other comments below and our extensive discussion of the study limitations). However, the same strategy enabled the generation of the genome-scale metabolic networks in the first place, and we added evidence that there is a high conservation of enzyme activators (see below, point 9 to this reviewer on page 12 & 13, and page 19 for reviewer #2).

Results: *"In the network of human enzyme inhibition interactions (Alam et al., 2017) all enzyme classes were equally susceptible to metabolic inhibitions. In contrast, in the activation network*

(Figure 1b) each enzyme class has a different prevalence of activated enzymes (Figure 1f). Out of the total intracellular activated enzymes, almost 36.33% belong to Transferases that is also the largest enzyme class of the Yeast9 metabolic network (Zhang et al., 2023), and are equally associated with all cell-intrinsic activated interactions (33.48%) (Figure 1f). Isomerases, and Ligases, which are the two smallest enzyme classes of the metabolic network, have even lower ratios of activated enzymes compared to non-activated enzymes (3.77%, and 8.43% respectively) (Figure 1f). In contrast, Oxidoreductases and Hydrolases, which are 2nd and 3rd largest metabolic enzyme classes, catalyzing thermodynamically favored metabolic reactions (non-equilibrium), have substantially higher percentage of intracellularly activated enzymes (19.47% and 21.51% respectively, Figure 1f)."

p5. What are the criteria for the degree of activation in Figure 2b? The authors should make their selection clear, including explicitly delineating and justifying them.

Response: With degree of activation, we referred to the frequency of activation interactions associated with enzymes or activators, represented by the number of edges connected to each node in the network (Figure 1b). For example, if an enzyme is activated by n metabolites, the degree of activation for that enzyme is n . Similarly, if a metabolite activates m different enzymes, the degree of activation for that metabolite is m . We have revised the methods section to include this explanation and clarified it in the main text within the results section.

Results - "The degree of activation for enzyme or activator metabolite nodes corresponds to the number of activating interactions, represented by the count of edges connected to each node."

Method - "Based on the frequency of activatory interactions, that is the number of edges for each node, the degree of activation for enzyme or activator metabolite nodes was calculated. For example, if an enzyme is activated by n metabolites, the degree of activation for that enzyme is n . Similarly, if a metabolite activates m different enzymes, the degree of activation for that metabolite is m ."

p6. The authors state that "most of the central metabolic pathways are significantly activating or activated by other pathways". The pathway analysis (as described in the Methodology section) simply counts the number of metabolites produced from one pathway that activates the enzymes of another pathway. Are the set of all metabolites in each pathway unique or do they overlap as substrates/products in other pathways as well? Related to this question is how do the authors take into account the fact that metabolites can occur in more than one pathway?

Response: We apologize for the lack of clarity regarding this analysis in our manuscript. In creating the pathway-to-pathway activation interactions network, we counted the number of metabolites produced in one pathway that serve to activate enzymes in another pathway, excluding any metabolites shared between the two pathways. Notably, the metabolites or enzymes under consideration were not confined to a single pathway; rather, if a metabolite

featured in multiple pathways, it was accounted for in all relevant pathways. Likewise, enzymes with involvement across multiple pathways were included in the analysis for each respective pathway. We have now clarified this in the method section. We have revised the methods section and provided more details to explain this clearly.

Method - "The pathway-pathway activation network was created based on the biochemical pathway definitions in the Yeast9 metabolic model (Zhang et al., 2023). First, for each pair of pathways, the activation frequency was calculated by counting the number of metabolites produced in one pathway that activated the enzymes of another pathway, excluding metabolites common to both pathways. This activation frequency was calculated for all pairs of metabolic pathways, resulting in a comprehensive table. Then, using a hypergeometric statistical test, the significance of activation between each pair of pathways was determined. Finally, the pathway-pathway activation network was reconstructed by linking two pathways if the P-value for one pathway activating another was less than 0.05."

p6. The author finds "In concordance with other studies, we find that the essential enzymes have..." but does not cite any studies. Please cite them. The author then states that essential enzymes have higher gene expression than non-essential enzyme Can the comparison be evaluated by comparing essential vs. non-essential enzymes on a pathway-to-pathway basis?

Response: We apologize for missing the citation to the appropriate place, we have now revised the text and appropriately cited the relevant studies. We also included an analysis of essential versus non-essential enzyme abundance.

Results: "It has previously been reported that essential enzymes are tightly regulated to maintain constant gene expression levels regardless of genetic or environmental perturbations (Grüning et al., 2010; Yang et al., 2012). Consequently, these genes are expected to exhibit less variation in gene expression (Park and Lehner, 2013). Consistent with those studies (Grüning et al., 2010; Park and Lehner, 2013; Yang et al., 2012), using microarray gene expression data across hundreds of conditions, we find that essential enzymes (whether activated or unactivated) have higher gene expression levels compared to non-essential enzymes (median log₂ gene expression 10.78 vs. 10.09, SD 1.65 vs 1.92, Supplementary Figure S3a). We did not observe any substantial difference in the gene expression levels between activated and non-activated enzymes, regardless of their essentiality (median log₂ gene expression 10.47 vs. 10.19, SD 1.84 vs 1.85, Supplementary Figure S3b)."

Response Figure 1 (Supplementary Figure S3 a,b in the manuscript). Gene expression variation between (a) essential and non-essential enzymes, and (b) activated and non-activated enzymes obtained from microarray gene expression experiments covering 665 unique experimental conditions. The gene expression values (log₂) across conditions were used to create the box plot.

Furthermore, we would like to thank the reviewer for their valuable advice to compare essential vs. non-essential enzymes on a pathway-to-pathway basis. We have performed this analysis, and the resulting figure (Response Figure 2) has been included in the supplementary information (Supplementary Figure S3c) and discussed in the main text.

Results - "We further conducted a pathway-specific analysis to assess gene expression variation between essential and non-essential enzymes. In line with the expression profile of overall enzymes, in the majority of pathways (34 out of 50), genes linked to essential enzymes demonstrate a higher median gene expression compared to those associated with nonessential enzymes (Supplementary Figure S3c)."

Response Figure 2 (Supplementary Figure S3c in the manuscript). **Pathway wise gene expression variation between essential and non-essential enzymes:** For each metabolic pathway, a set of essential and non-essential enzymes were identified using *in silico* enzyme knockout experiments and gene expression variation between essential and non-essential enzymes was calculated using the public data obtained from microarray gene expression experiments.

p6. The author state that "NAD biosynthesis is activated by two different pathways" but does not provide further insight on how this happens. In general, this lack of insight, and instead a rote listing of results, (both here and elsewhere) is a key detriment to the impact of the manuscript.

Response: We thank the reviewer for their comment. As we have used the most recent metabolic model as the basis of this study, the entire text has been revised. This particular piece of statement is no longer present in the revised manuscript. However, we understand the importance of detailed explanation of the result, therefore, we have added more insights of our results and described them in both results as well as the discussion sections.

Results section:

"Next, we examined the extent to which different biochemical pathways activate one another (Figure 2d). Among the 72 metabolic pathways in the Yeast9 network, only half (36, or 50%) show significant activation interactions with other pathways (hypergeometric test; p value <0.05). The remaining pathways, although containing activated enzymes, neither significantly activate nor are significantly activated by other pathways (hypergeometric test; p value <0.05). Of the pathways with activation interactions, the majority (26 out of 36, or 72%) either activate or

are activated by other pathways. Thirteen pathways activate only a single pathway, 17 are activated by just one pathway, while only 4 of the 36 pathways exhibit significant self-activation, highlighting the general specificity of activation interactions. This notable degree of trans-activation between pathways reflects the interconnected nature of cellular metabolism. It ensures that the metabolic pathways are specifically activated and coordinated to dynamically adapt in order to meet the cellular needs.

Additionally, only 6 pathways are activated by more than one pathway, revealed the enrichment analysis (hypergeometric test; p value <0.05). One notable pathway that is significantly activated by the largest number of pathways (8 in total) is pyruvate metabolism. This pathway is known to be highly regulated because it serves as a key metabolic hub, linking glucose breakdown to various cellular processes. Pyruvate metabolism acts as a bridge between glycolysis and several major biochemical processes, including the citric acid cycle, anaerobic fermentation, and anabolic pathways for synthesizing amino acids, depending on cellular needs. Similarly, due to the broad role of Histidine metabolism in protein synthesis, energy production, nitrogen regulation, and stress response, and its dynamic regulation according to cellular need, it is also activated by many specific pathways. There are some pathways including Fatty acid ester pathway, Glutathione metabolism, Arginine biosynthesis, Glycosylphosphatidylinositol (gpi)-anchor biosynthesis, and Nicotinate and nicotinamide metabolism which act as hub activators, activating more than one pathways significantly.”

p7. Standard deviations are not reported for all averages of gene expression. What is the measurement error on the average gene expression levels and are the differences significant? If the standard deviations that are not reported are similar to those that are, then the differences cannot be considered significant.

Response: Thank you for your suggestions. We have now incorporated standard deviation into our analysis, and these values are included in both the main text and the figures (Response Figure 3). Additionally, we would like to clarify that we evaluated differences in gene expression values, whether significant or non-significant, using T-test statistics. This test takes into account both the mean and standard deviation in its calculations. Differences were considered statistically significant if the p -value obtained from the T-test was less than 0.05.

“We did not observe any substantial difference in the gene expression levels between activated and non-activated enzymes, regardless of their essentiality (median \log_2 gene expression 10.47 vs. 10.19, SD 1.84 vs 1.85, Supplementary Figure S3b). However, enzymes that are essential for growth and are intracellularly activated exhibit the highest gene expression values and the least variance (median \log_2 gene expression 10.87, SD 1.67, Figure 3c iv). This indicates that these essential activated enzymes are consistently upregulated and maintain the highest expression levels. In contrast, enzymes that are non-essential for growth and rely on activation for their expression show the lowest median gene expression across conditions and the highest variance (median \log_2 gene expression 9.84, SD 2.02, Figure 3c iv). These enzymes are infrequently expressed and may not be tightly regulated. However, for the non-activated

enzymes we did not observe major differences between essential and non-essential enzymes (median log₂ gene expression 10.18 vs 9.92, SD 1.81 vs 1.95, Figure 3c iv).”

(iv)

gene expression variation

Response Figure 3 (Figure 3c iv in the manuscript). **Gene expression variation** among activated essential, activated non-essential, non-activated essential, and non-activated non-essential enzymes. In the box plot, microarray gene expression experiments, covering 665 unique experimental conditions, were used to obtain the processed gene expression values (log₂) across conditions.

p7. In the results reported on the previous page the author comes to the conclusion by comparing the average gene expression. However, even if one ignores the statistical matter, it is not clear how gene expression level can be used here, as differences in translation can result in different protein levels. The authors should clearly justify their use of expression data and raise all the limitations that brings. Are there protein level data that could better support the hypotheses? The manuscript would be substantially improved by their incorporation.

Response: We appreciate the reviewer’s insightful observation. In our analysis, we had used mRNA expression data, understanding that differences in translation efficiency can indeed lead to discrepancies between mRNA and protein abundance. Following the advice of the reviewer, we have additionally performed the analysis based on the protein expression datasets across the proteome profile on thousands of knockout strains as measured by (Messner et al., 2023a). We have examined the proteome profile of essential, non-essential, activated and non-activated enzymes. Our results show similar observations for both proteome and transcriptome datasets.

Results section:

“Similarly, we examined protein expression variations among essential, non-essential, activated, and non-activated enzymes using proteome profiles from thousands of knockout strains, as measured by liquid chromatography tandem mass spectrometry (Messner et al., 2023b). Our findings show consistent expression patterns in both proteome and transcriptome datasets (Supplementary Figure S4).”

Response Figure 4 (Supplementary Figure S4): The protein expression variation among activated essential enzymes, activated non-essential enzymes, non-activated essential enzymes, and non-activated non-essential enzymes.

p7. Although the authors do remove highly connected co-factors such as ATP or NAD/NADPH, etc. from the graph with nodes representing metabolites connected by chemical reactions contain, but that fact is buried in the methods and should be mentioned. Also, in the methods the phrase "Some of the highly connected metabolites including..." occurs but for reproducibility all of the excluded metabolites should be explicitly listed. How did the authors define a "highly connected metabolite"?

Response: We appreciate the reviewer's valuable input. To refine our network analysis for shortest path length calculation, we excluded highly connected metabolites—those with over 50 connections across any cellular compartment in the metabolic network graph, mostly including cofactors. This step aimed to mitigate undue connections arising solely from such highly connected metabolites, like cofactors. The list of these metabolites is provided in the Supplementary File 1. We have incorporated this clarification into both the method section and highlighted it within the results section.

Results - "First, we converted the metabolic network model into a graph where nodes represent metabolites and edges are established when two metabolites are connected by a chemical reaction. We removed 25 highly connected metabolites (metabolites, mostly cofactors, with more than 50 metabolic connections in any cellular compartment; Supplementary File 1) in order to avoid misleading connections."

Method - "Then, highly connected metabolites, having more than 50 metabolic interactions in any single cellular compartment, were removed (list of removed metabolites are in Supplementary File 1)."

p8. Once again, nebulous "other studies" are mentioned but not cited, and the list is reported as "well-characterized" without defining what that means. One of the studies the authors refer to is probably Noor et al. 2010 (10.1016/j.molcel.2010.08.031) but please actually cite all the studies referred to. Moreover, the identity of these 12 metabolites and how they differ (presumably, since the authors state "similar to this overservation") and the implications thereof.

Response: The reviewer is correct; we intended to reference Noor et al. (2010), and we apologize for the omission. This citation has now been included. In the results section, we emphasize the optimality of producing activator metabolites within the metabolic network, demonstrating that activators have shorter path lengths from glucose molecules compared to non-activator metabolites. We further note that this inherent optimality is not exclusive to activators but is a fundamental feature of metabolic systems. This is supported by Noor et al. (2010), who showed that only 12 key precursor metabolites form the foundation of cellular biomass. The main text has been revised accordingly.

Results: "These results suggest an inherent optimality in the production of cellular activator metabolites within the metabolic network. This indicates that intracellular activators begin to form shortly after nutrient uptake, subsequently driving overall metabolism in a positive direction. Furthermore, this optimality is not unique to activators but represents a fundamental feature of metabolic systems. This is demonstrated by the finding that only 12 key precursor metabolites, located at various branch point nodes and converted from carbon source nutrients via the shortest path lengths, form the foundation of cellular biomass (Noor et al., 2010)."

p8. The authors use graph theory to look at the shortest path for activators and simply state that "majority of the activator metabolites have the shortest path length of 5 whereas for non-activator this distance ranges from 5-7". How many are these exactly out of how many (that is, give the result as a non-reduced fraction or ratio).

Response: We transformed the Yeast9 metabolic model into a graph and calculated the shortest path length from the glucose molecule for each metabolite. The resulting shortest path lengths distribution for both activators and non-activators were visualized using a violin plot, which illustrates the density of data distribution. In Figure 4a, depicting the shortest path length

distribution for both groups, we observed that activators primarily exhibit a peak at the value of shortest path length 6 while non-activators show peak at value 8. To enhance clarity, we have revised the text and included median values in the main text. Additionally, a small box plot is also incorporated within the violin plot.

Results - "The bulk of activator metabolites exhibit a shortest path length of 6, contrasting with non-activators which is more than 8 (with mean, median and standard deviation of shortest path lengths for activators at 6.29, 6 and 1.71 respectively, and for non-activators at 7.99, 8 and 1.54, Figure 4 a,b)."

Is this shortest path accounting for reactions that only produces intermediate metabolites not used elsewhere in the system (e.g. aconitase) or the presence of multienzyme complexes that are modeled with several reactions (e.g., fatty acid synthase)?

Response: We included all metabolic reactions in constructing the network graph. However, we excluded the highly connected metabolic cofactors from the metabolic network. If the product of a reaction is not used elsewhere in the metabolic network, then such a metabolite becomes dead end, which is not considered when calculating the shortest path length. Furthermore, if there are gaps in the network for any metabolite, we cannot calculate the shortest path length for those compounds either, so they were disregarded from the analysis. We have reworked the methods section, to make this clearer.

p8. The authors very briefly bring in an analysis of microarray gene expression data from the ArrayExpress database. However, no mention of controlling for transcription level regulation is mentioned in the methods. How are the authors confident that what is observed can be attributed solely to allosteric regulation? The case for their conclusions is not sufficiently made and the authors should reevaluate the results after taking into account transcription regulation. Moreover, protein levels do not exactly correlate with gene expression data, and the authors do not bring up this caveat.

Response: We apologize for the confusion. In our analysis, we did not examine the regulatory controls to expression values; we have used the mRNA (and now also proteomics data by Messner et al, 2023) to assess abundance and abundance variation in the enzymes, but did not study how allostery and gene expression regulation interact. This is an intriguing topic, but out of scope for a single manuscript. We have added a caveat, stating that we are not studying gene expression herein, but now Reference our previous work in which we came to the conclusion, that about 50% of metabolite concentration variation is explained by enzyme abundance changes (Zelezniak et al. 2018)

Results: "It is important to note that we are not studying metabolic regulation through mRNA or protein expression; rather, we are using expression variation data to detect differences between essential and nonessential enzymes, as well as activated and unactivated enzymes, since

approximately 50% of the variation in metabolite concentrations can be attributed to changes in enzyme abundance (Zelezniak et al., 2018)."

p9. The authors assume as ground truth that all interactions given in BRENDA are present in their network. The authors do correctly note in the discussion that they might (daresay are) overestimating the number of activators. Was any attempt made to examine yeast (or even better, *S. cerevisiae*) studies to validate or curate the information from BRENDA? Were proteins across all clades of life used? Another avenue could be to use protein structures (predicted or measured) to evaluate if there are binding sites for the metabolites.

Response: Please see our comment above (page 2, page 13). We are more than aware of this limitation, but it was the best choice available for this study, given the species-specific data is still too sketchy to create a network wide overview. We would like to highlight that indeed the genome-scale reconstruction of the metabolic network is, at the end of the day, also based on accrued cross-species data, which enabled the construction of these networks in the first place. Having said that, we agree and recognize that the inclusion of yeast-specific data would improve the accuracy of our network. We have thus gathered yeast specific data and come to similar conclusions. For example, see the following Results paragraph. Looking ahead, advances in structural biology and high-throughput experimental techniques may allow for more precise mapping of these interactions, ultimately leading to the development of species-specific networks. For now, our approach lays a valuable foundation for investigating enzyme activity interactions, with the understanding that further refinement will be possible as more detailed and specific data becomes available. We have clarified this in the result section.

Result

*"To construct a genome-scale enzyme-metabolite activation interactions, we obtained the topology of the *Saccharomyces cerevisiae* metabolic network, and enriched it with cross-species acquired data of enzyme activation. For each metabolic enzyme (in total 635 enzymes) in the genome-scale metabolic model (Yeast9) (Zhang et al., 2023) a list of all associated activatory molecules were downloaded from the Brenda database, using SOAP clients instructions (Chang et al., 2021). The BRENDA database contains the evidence of activation interactions for 140 metabolic enzymes of which 88 enzymes (63%) have at least one activation interaction with the same activator molecule in other species, supporting our assumption that these activation interactions are conserved across species."*

p12. The author states in that activator information is directly pulled from BRENDA using a method akin to a web scraping tool (SOAP). The BRENDA database often does not accurately provide sufficient information for curating allosteric information as they often depend on growth conditions. Did the authors check with each activator used in the network model and verify the activator metabolite for the given experimental condition? For the 655 unique datasets examined, were filters added to distinguish aerobic vs

anaerobic conditions? The authors should clearly comment on in the manuscript on how much of their network and results are condition dependent.

Response: We thank the reviewer for making this point. They are correct that for a majority of competitive and allosteric regulators in BRENDA, but indeed also the primary literature, does not have information to which degree they are condition dependent. Therefore, our network does not reach a condition-specific resolution. There is nothing we could do about it, simply as the primary experimental data is not there. We have added to the discussion that our study makes it obvious that we, as a field, do not sufficiently understand the condition-dependency of enzyme activators. We hope that our study stimulates more experimental work in this direction.

Discussion

*“It is important to note that our approach operates under the fundamental assumption that metabolic activatory interactions are conserved across species (Guan et al., 2020; Ribeiro et al., 2020). This is a necessary assumption for our study to achieve sufficient network coverage; no activator network is currently sufficiently mapped at the single species level. While this assumption certainly has limitations (for example, our approach might overestimate the total number of activated interactions by an unknown degree), there is indeed evidence that many activation reactions are indeed conserved. For example, 63% of *S. cerevisiae* metabolic enzymes share at least one activation interaction with other species. Additionally, enzyme regulations in BRENDA and the literature do not provide enough information to determine the extent of their condition dependency, limiting our network's ability to achieve condition-specific resolution.”*

Minor points:

p1. Affiliation 5 is labeled with the number 2 in the author list.

Response: We have fixed that typo.

p2. The authors should consider turning on line numbers.

Response: We have inserted the line numbers.

p4. The method for downloading BRENDA (i.e., SOAP) is listed in the Results section but not mentioned in Methodologies. Since it is a method, this detail should be moved to Methodologies.

Response: Thank you for pointing this out. We did mention the details of making the network in the method section, however, based on the reviewer's request we have elaborated the method section, and more details were added.

p6. The authors states "several central metabolic pathways (15/18) are..." but does not define which 18 pathways are classified as the central metabolic pathways. Please define them in the methods section or include them in supplemental information.

Response: By central metabolism we meant non transport metabolic pathways. In order to avoid further confusion, we have revised the text and changed it to metabolic pathways.

p6. The authors use the term 'hub activator' and the wording suggests that they have defined it as a node that has activation connections with more than 15 enzymes but the sentence should be rephrased to make that clear. Also, how was the cutoff of 15 selected? Do the results change if a number such as 12 or 20 or top 10% is used?

Response: The reviewer is correct - we worked with a common definition in the field which considers nodes 'hubs' if they have a high number of connections (we used a cut off of 15 in our study, which we admit has an arbitrary component). Based on the reviewer's suggestion, we have also examined the nodes [activators and enzymes] with different numbers of connections (12, 15, and 20) and the results give the same conclusions (Response Figure 5, Supplementary Figure S2). We have also added a clear definition in our results section.

(a) Degree of activation for highly activating hub metabolites

(b) Degree of activation for highly activated hub enzymes

Response Fig 5 (Supplementary Figure S2 in the manuscript) (a) The degree distribution of activator metabolites within the cell-intrinsic activation interaction network reveals that highly interactive activators (degree > 12, degree > 15, degree > 20) are mostly essential for growth. (b) The degree distribution of activated enzymes within the cell-intrinsic activation interaction network reveals that highly activated enzymes (degree > 12, degree > 15, degree > 20) are mostly non-essential for growth.

p7. How does the non-activator distance range from 5-7 if the reported maximum length is 6?

Response: We apologize for this typo. The results have now changed, and we have calculated the shortest path length and written correctly in the revised manuscript.

Results:

“The bulk of activator metabolites exhibit a shortest path length of almost 6, contrasting with non-activators which is almost 8 (with mean, median and standard deviation of shortest path lengths for activators at 6.29, 6 and 1.71 respectively, and for non-activators at 7.99, 8 and 1.54, Figure 4 a,b).”

p8. The authors state "Figure 4a shows a subnetwork of the main cell-intrinsic activation interaction" however the reference should probably be Figure 5a since Figure 4a does not show a subnetwork but Figure 5a does.

Response: Thank you for pointing this out. We have now corrected this.

p9. Human metabolism is again mentioned yet its relevance is unclear.

Response: Please see the response to comment #p2 on page 2 where we have addressed this comment in detail. The other study used the human rather than the yeast topology as a basis, but the general properties of enzyme inhibitors, as addressed in this study, are conserved across species.

Reviewer #1 (Significance (Required)):

The manuscript would require significant changes before acceptable for publication.

We thank the reviewer for their positive assessment of our work, as well as their valuable suggestions and comments. We have revised the manuscript significantly, incorporating the reviewer's suggestions and addressing all their comments. We hope our responses meet their expectations.

Reviewer #2 (Evidence, reproducibility and clarity (Required)):

The authors generate and analyse a genome-scale metabolic network of allosteric activation in *S. cerevisiae*. This is a very interesting and relevant piece of work that sheds light upon the role of allosteric interactions, in particular allosteric activation, in the regulation of metabolism.

Response: Thank you for your thoughtful and positive feedback on our work. We are delighted to hear that you found our study on cellular activation interactions within the metabolic system both interesting and relevant. Cellular activations interactions, especially allosteric activation, plays a crucial role in the complex regulation of metabolism, and we are pleased that our research has contributed to a better understanding of this important aspect. We have carefully reviewed all your concerns and have addressed them in the revised manuscript. Our detailed point-by-point responses are provided below.

Major issues:

- One major problem with allosteric regulation is that we still do not have an unbiased high-throughput method to completely cover the whole enzyme-metabolite combinatorial space. Entries in databases like BRENDA correspond to in vitro studies that are biased towards well-studied pathways. Therefore, it is not surprising that core pathways like

central carbon, nucleotide, and amino acid metabolism seem to be the mostly regulated. The authors should at least discuss this issue.

Response: We totally agree with the Reviewer that we do not have an unbiased high-throughput method to completely cover the whole enzyme-metabolite combinatorial space, and this is precisely the reason why we accumulate the century of knowledge of individual biochemical experiments as collected in BRENDA, to approximate the activator network.

We also agree that central pathways such as glycolysis, the pentose phosphate pathway and the TCA cycle are more studied, and that there might be a literature bias. However, the more complex regulation of these pathways is not just explained by a literature bias. They are most connected in the metabolic network, as many other pathways branch off, the enzymes of these pathways are among the most abundant and most regulated enzymes also in an unbiased proteome investigation (Messner et al., 2023a), and our recent study using AlphaFold predicted structures across the metabolic network, shows that they are the structurally most constrained, also in the non-catalytic, regulatory regions and the surface (Lemke et al., 2024). Thus, also in unbiased, genome-scale datasets, glycolysis, TCA cycle and PPP come out among the most central metabolic hubs. We have added this point to the result section.

- Additionally, the authors could try to mitigate this issue by normalizing the number of reported allosteric interactions for a metabolite or an enzyme by either the corresponding number of literature references in BRENDA for the same metabolite/enzyme or by the number of reported inhibitory interactions. Are there metabolites that are frequent activators but not frequent inhibitors? Or reactions that are frequently activated but not frequently inhibited?

Response: This a thoughtful idea by the reviewer, but in trying this, we have detected a negative correlation between literature citations and the number of allosteric regulators. This actually shows that the signal is much stronger than the literature bias- TCA cycle and glycolysis are indeed more likely regulated by enzyme activators, compared to peripheral pathways. Please see also the previous paragraph.

- Enzyme essentiality was computed on glucose minimal medium, which is an artificial laboratory setting and not a realistic scenario for species living in nature. Therefore, there is no distinction between conditionally essential enzymes and the universally essential enzymes (those that would also be essential in a complete medium). I think it would be important to make this distinction and test if it has any effect on the results presented.

Response: The reviewer is correct, and the best reference in this context is perhaps the seminal work of Maureen Hillenmeyer et al, which have shown that every yeast gene is essential in at least some conditions (Hillenmeyer et al., 2008). As a consequence, being essential and non-essential indeed is a quantitative rather than a binary question; some genes

are essential all the time, but all the rest are conditionally essential. However, the growth in laboratory standard conditions remains a helpful approximation to identify most of the 'universally essential genes', as the lab conditions are not stressful, not toxic, and not metabolically challenging, and thus come close to the 'minimal set' of essential genes. Also, for a species that often grows on fruits, grapes or in the gut, high glucose concentrations are physiological. Indeed, the genes which are essential under these favorable lab conditions are also essential under most other conditions as well, are in average more abundant (Messner et al., 2023a), and more connected than the other 'conditionally essential' genes, and thus form a useful gene group also for our study. In order to address the valuable point by the reviewer, we have refined the definition, and clearly state that 'non-essential' refers to the conditionally essential genes not essential under lab conditions.

Result:

"The essentiality of enzymes and metabolites, predicted using the flux balance analysis approach for optimal growth in glucose minimal media, was integrated with the cell-intrinsic activation network (Figure 1b). An enzyme or metabolite is deemed essential if the Yeast9 metabolic model fails to produce biomass in glucose minimal media (standard laboratory condition) after all reactions associated with that enzyme or metabolite are removed. Otherwise, the enzyme or metabolite is considered non-essential, though they may be conditionally essential under different conditions."

- Also, I wonder why authors did not simply use in vivo essentiality data (which can be found in the literature for such a well-studied organism like this one) instead of predicted in silico data.

Response: We appreciate the reviewer's observation regarding the availability of gene essentiality experimental data for well-studied organisms such as *Saccharomyces cerevisiae*. It is important to emphasize that gene essentiality predictions show strong alignment with experimental results, as reported in several studies (O'Brien et al., 2015). To address the reviewer's concern, we conducted in silico gene knockout experiments in minimal media and compared the predicted essential genes with experimental data from our previous study under the same conditions (Aulakh et al., 2023), finding an 83.3% agreement between predictions and experiments. Furthermore, when comparing our predicted essential genes in minimal media with experimentally verified essential genes across different conditions (Cherry et al., 2012; Mülleder et al., 2016), we observed a 75% agreement.

However, in this study, our focus is on the essentiality of enzymes and metabolites rather than genes. Currently, there is limited experimental evidence available for enzyme knockout experiments.

- In the first paragraph of page 7 the authors describe the differences in gene expression observed in Figure 3c(iii). I would argue that in fact the conclusion is that there are no observable differences. It is important to note that magnitude of difference and statistical

significance are different things. These differences might be statistically significant (due to large sample size), but they are of very small magnitude. One can conclude, with statistical confidence, that there is almost no difference.

Response: We thank the reviewer for their comment. We agree that the magnitude of difference for all comparison is not very high, however, there is a significant difference between activated-essential and activated-nonessential enzymes (median expression 10.87 vs 10.24), similarly non activated-essential and non activated-nonessential enzymes also exhibit a significant difference in the overall gene expression levels (median expression 10.66 vs 9.96)

- I am a bit skeptic of the results regarding pathway length. How is it possible that most metabolites are on average only 5 steps away from glucose, and most enzymes on average only 4 steps away? I would expect most metabolites and enzymes to be tens of steps away. For instance, the aromatic amino acid pathway is several steps long, how would tryptophan be only 5 steps away from glucose? I suspect there are several shortcuts due to the presence of co-factors in the graph generating algorithm. The authors say they remove "some of the highly connected metabolites like h, h₂o, adp, atp, amp, pi, nad, nadh". This part needs a more rigorous description. All cofactors (including redox pairs, CoA groups, etc) should be removed.

Response: We would like to thank the reviewer for their feedback. We have revised this analysis and all highly connected metabolites such as cofactors, coA have been removed from this analysis to calculate the more realistic connection length between other cellular metabolites and glucose. The metabolic network follows the definition of a *small-world network*, where nodes are connected to each other in maximum 5-6 steps (i.e. Six degrees of separation). Moreover, in supplementary file 1, we have listed all the highly connected metabolites which were removed for calculating the shortest path length.

- The authors mention that they used all allosteric activations in BRENDA, not only those reported for *S. cerevisiae*, assuming that allosteric regulation is conserved. In fact, there are some articles arguing that allosteric regulation might not be under strong selective pressure, and therefore might not be as conserved as we might think. I would strongly recommend building a network using only interactions reported at the same species (or genus, or any relatively conserved taxonomic) level, and re-running some of the tests just to confirm that they don't affect the main conclusions.

Response: As aforementioned (page 2, 12 and 13 in response to reviewer #1) the reason we, and previous studies of similar kind, had to go across species in the first place was that for any single species, there was not sufficient data to generate a sufficiently well-covered network necessary for deriving robust, general conclusions We would like to highlight that Brenda already accumulated a century of enzymology work in more than 157000 primary papers, so this situation will not change soon unfortunately. It's the best we can do, and we have extensively highlighted this caveat in the paper. Having said that, we believe our choice is a

sensible strategy. As a sanity check for the conservation of activators, we took all activated enzymes of *S. cerevisiae*, reported in the BRENDA database, and queried how many of these are activated by the same activating molecule in other species. We find that 63% of the enzymes found in yeast have common activators in other species. This result gives reasonable confidence that most activators are not species specific. We have added this information to the manuscript.

Results:

“The BRENDA database contains the evidence of activation interactions for 140 metabolic enzymes of which 88 enzymes (63%) have at least one activation interaction with the same activator molecule in other species, supporting our assumption that these activation interactions are conserved across species.”

- It would be interesting to include in the discussion some comments about why evolution might favor allosteric activation instead of transcriptional regulation to increase enzyme activity. On one hand, allosteric activation is faster, and would allow the cell to respond more rapidly to environmental changes. On the other hand, producing enzymes which are unused and waiting for activation is more costly. There are many recent papers showing how protein production cost is the main metabolic constraint limiting growth. It would be interesting to compare the degree of enzyme regulation with the respective protein cost.

Response: These are reasonable and interesting discussion points, and we agree with them. We have expanded the discussion accordingly. We believe the main argument are indeed a) speed - allostery regulates metabolism orders of magnitude faster than hierarchical regulation - and b) its cost efficiency. To compare activation with cost is effectively covered in our manuscript, as we detect a relationship with enzyme abundance (Response Figure 4 (Supplementary Figure S4)), which overall correlates reasonably well with cost. We have also added this point to the manuscript.

- In the discussion section, please replace all the EC numbers and KEGG ids listed with enzyme and metabolite names. In fact, it would be nice to have a table in the main text with the top 10 activator metabolites and top 10 activated enzymes.

Response: We have revised the text and added enzyme and metabolite names together with IDs. Also, we have listed the top 10 most activated enzymes and activating metabolites in Table 1.

Table 1: Top 10 intracellular activating metabolites and activated enzymes.

Top 10 cellular activating metabolites	Top 10 cellular activated enzymes
--	-----------------------------------

Activator s ID	Activators name	# activa tion intera ctions	ess enti ality	EC	Enzyme Name	# activat ion intera ctions	essen tiality
C00097	L-Cysteine	67	Y	EC:1.4. 1.2	glutamate dehydrogen ase	28	N
C00002	ATP	61	Y	EC:2.7. 1.40	pyruvate kinase	28	Y
C00051	Glutathione	59	N	EC:3.5. 4.1	cytosine deaminase	25	N
C00009	Orthophos phate	43	Y	EC:3.1. 3.11	fructose- bisphospha tase	24	N
C00469	Ethanol	36	N	EC:3.1. 3.2	glyceropho sphatase	23	N
C00157	Phosphatid ylcholine	27	Y	EC:3.1. 4.4	phospholip ase D	21	N
C00008	ADP	27	Y	EC:3.1. 2.2	β -D- glucosides	19	Y
C00020	AMP	24	Y	EC:3.1. 2.1	acetyl-CoA hydrolase	18	N
C00750	Spermine	24	Y	EC:2.4. 1.11	Glycogen synthase	17	Y
C00350	Phosphatid ylethanol amine	24	Y	EC:2.7. 1.11	6- phosphofru ctokinase	17	N

- I am curious why the authors used the outdated iMM904 model (Mo et al, 2009) and not a more recent model like Yeast8 (Lu et al, 2019).

Response: We thank the reviewer for their comments on the use of an outdated model. The other reviewer has also pointed out this important issue. We have updated the entire study by using the most recent mathematical model as the basis of this study. Please see our response to the Reviewer #1 comment, page 2-3.

- Please provide a repository link with all the code and data so that readers can try to reproduce your results.

Response: All the relevant data is provided in the supplementary and explained in detail in the method section. We have added an additional comment regarding the scripts which is available upon request.

****Referees cross-commenting****

It seems that there are some points in which all reviewers are in agreement, especially when it comes to improve the methodology description, such as the download of non-yeast data from BRENDA, the use of an outdated yeast model, and a more clear definition of pathways and pathway distance.

Nonetheless, I think it is quite a nice and relevant piece of work. I look forward to see it published.

Response: We thank the reviewers for their valuable suggestions which have helped in improving the quality of the paper. We have carefully considered their concerns and addressed them. Most of the comments were resolved by detailed reanalysis and several new analyses.

All referees and the editor raised valid concerns about our use of an outdated metabolic model to examine activation interactions. We agree with these concerns and have updated the study using the latest metabolic model (Yeast9). As a result, we have fully revised the manuscript, including significant changes to the text and figures. However, I am pleased to report that while the text and numerical values have changed, the core message of the paper remains intact. We have also provided a clearer definition of pathways and added more details about the distance between enzymes and activators from glucose nutrients.

Lastly, we maintained the inclusion of non-yeast data in the network reconstruction, as we are confident in the correctness of our approach. We have addressed this in our response to the relevant comment.

Reviewer #2 (Significance (Required)):

I find this study to be of high significance. It explores an often overlooked aspect of metabolic regulation.

Response: On behalf of all co-authors, I would like to thank the reviewer for their positive feedback on our work. We are pleased that the reviewer shares our enthusiasm for highlighting the often-overlooked aspects of metabolic regulation.

Reviewer #3 (Evidence, reproducibility and clarity (Required)):

Zubaida et al mapped known information of metabolic activators onto a genome scale metabolic model to address the importance of activation and its locality. I found the study to be excellent and clear. I do not feel that reproducibility is a criteria that can easily be addressed in such studies.

We thank the reviewer for their positive assessment of our work. We have incorporated the suggestions and addressed the concerns in the main text. Our point-by-point response is below

I only have three minor issues which I address in order of importance.

1) the authors should clarify if the activator information they use from BRENDA is for yeast specifically or merely in ANY organism. Similarly was it collected in physiologically relevant conditions or at the enzymes optimal conditions. I feel this information should be provided as they may effect the outcome somewhat. The latter cannot be helped as it was how enzymology typically was done but this should be discussed.

Response: We apologize for not clearly explaining this part in the manuscript as all reviewers have asked a similar question. Please see our response on page 2, 12, 13 and 19. We have used the activatory interactions information across species and they are not yeast specific. We have added the below text in the manuscript and also highlighted this in other places.

Result

*Notably, we used metabolic enzymes from *Saccharomyces cerevisiae* as the foundation for the cell-intrinsic activation network, while evidence for enzyme activation was gathered from multiple species. Among the total metabolic enzymes of *S. cerevisiae*, activation interactions have been reported for 140 enzymes in the BRENDA database. Of these, 88 enzymes (63%) have at least one activation interaction with the same activator molecule in other species, supporting our assumption that a large fraction of activation interactions are conserved across multiple species.*

2) the authors talk about activation by extracellular molecules I assume that they are implying that these are physiologically irrelevant however this should perhaps be explicitly stated.

Response: We are sorry for this confusion. We are not saying that activation of an enzyme by extracellular molecules is physiologically irrelevant. We have ignored all extracellular activator molecules from our study as we aimed to create the cell intrinsic activatory interactions network where only cellular activating metabolites were considered.

3) at many times metabolites start with an uppercase letter this is only correct when they are also the first word of a sentence.

Response: We thank the reviewer for pointing this out. We have revised the manuscript and these types were corrected.

Reviewer #3 (Significance (Required)):

This is a highly important study with broad implications regarding the regulation of metabolism. I found it clearly worded, well planned and well executed.

We once again thank the reviewer for their positive remarks, and we hope the revisions and responses have improved the manuscript to their satisfaction

Response References:

- Alam MT, Olin-Sandoval V, Stincone A, Keller MA, Zelezniak A, Luisi BF, Ralser M. 2017. The self-inhibitory nature of metabolic networks and its alleviation through compartmentalization. *Nat Commun* **8**:16018.
- Aulakh SK, Sellés Vidal L, South EJ, Peng H, Varma SJ, Herrera-Dominguez L, Ralser M, Ledesma-Amaro R. 2023. Spontaneously established syntrophic yeast communities improve bioproduction. *Nat Chem Biol* **19**:951–961.
- Chang A, Jeske L, Ulbrich S, Hofmann J, Koblitz J, Schomburg I, Neumann-Schaal M, Jahn D, Schomburg D. 2021. BRENDA, the ELIXIR core data resource in 2021: new developments and updates. *Nucleic Acids Res* **49**:D498–D508.
- Cherry JM, Hong EL, Amundsen C, Balakrishnan R, Binkley G, Chan ET, Christie KR, Costanzo MC, Dwight SS, Engel SR, Fisk DG, Hirschman JE, Hitz BC, Karra K, Krieger CJ, Miyasato SR, Nash RS, Park J, Skrzypek MS, Simison M, Weng S, Wong ED. 2012. Saccharomyces Genome Database: the genomics resource of budding yeast. *Nucleic Acids Res* **40**.
- Grüning N-M, Lehrach H, Ralser M. 2010. Regulatory crosstalk of the metabolic network. *Trends Biochem Sci* **35**:220–227.
- Guan X, Upadhyay A, Chakrabarti R. 2020. Mechanism-based enzyme activating compounds. *bioRxiv*. doi:10.1101/2020.04.08.032235
- Hillenmeyer ME, Fung E, Wildenhain J, Pierce SE, Hoon S, Lee W, Proctor M, St Onge RP, Tyers M, Koller D, Altman RB, Davis RW, Nislow C, Giaever G. 2008. The chemical genomic portrait of yeast: uncovering a phenotype for all genes. *Science* **320**:362–365.
- Lemke O, Heineike BM, Viknander S, Cohen N, Steenwyk JL, Spranger L, Li F, Agostini F, Lee CT, Aulakh SK, Nielsen J, Rokas A, Berman J, Zelezniak A, Gossmann TI, Ralser M. 2024. The Role of Metabolism in Shaping Enzyme Structures Over 400 Million Years of Evolution. *bioRxiv*. doi:10.1101/2024.05.27.596037
- Messner CB, Demichev V, Muenzner J, Aulakh SK, Barthel N, Röhl A, Herrera-Domínguez L, Egger A-S, Kamrad S, Hou J, Tan G, Lemke O, Calvani E, Szyrwił L, Mülleder M, Lilley KS, Boone C, Kustatscher G, Ralser M. 2023a. The proteomic landscape of genome-wide genetic perturbations. *Cell* **186**:2018–2034.e21.
- Messner CB, Demichev V, Muenzner J, Aulakh SK, Barthel N, Röhl A, Herrera-Domínguez L, Egger A-S, Kamrad S, Hou J, Tan G, Lemke O, Calvani E, Szyrwił L, Mülleder M, Lilley KS, Boone C, Kustatscher G, Ralser M. 2023b. The proteomic landscape of genome-wide genetic perturbations. *Cell* **186**:2018–2034.e21.
- Mülleder M, Calvani E, Alam MT, Wang RK, Eckerstorfer F, Zelezniak A, Ralser M. 2016. Functional Metabolomics Describes the Yeast Biosynthetic Regulome. *Cell* **167**:553–565.e12.
- Noor {elad, Eden E, Milo R, Alon} U. 2010. Central Carbon Metabolism as a Minimal Biochemical Walk between Precursors for Biomass and Energy. *Mol Cell* **39**:809–820.
- O'Brien EJ, Monk JM, Palsson BO. 2015. Using Genome-scale Models to Predict Biological Capabilities. *Cell* **161**:971–987.
- Park S, Lehner B. 2013. Epigenetic epistatic interactions constrain the evolution of gene expression. *Mol Syst Biol* **9**:645.
- Ribeiro AJM, Tyzack JD, Borkakoti N, Holliday GL, Thornton JM. 2020. A global analysis of function and conservation of catalytic residues in enzymes. *J Biol Chem* **295**:314–324.

- Yang J-S, Seo SW, Jang S, Jung GY, Kim S. 2012. Rational Engineering of Enzyme Allosteric Regulation through Sequence Evolution Analysis. *PLoS Comput Biol* **8**:e1002612.
- Zelezniak A, Vowinckel J, Capuano F, Messner CB, Demichev V, Polowsky N, Müllender M, Kamrad S, Klaus B, Keller MA, Ralser M. 2018. Machine Learning Predicts the Yeast Metabolome from the Quantitative Proteome of Kinase Knockouts. *Cell Syst* **7**:269–283.e6.
- Zhang C, Sánchez BJ, Li F, Eiden CWQ, Scott WT, Liebal UW, Blank LM, Mengers HG, Anton M, Rangel AT, Mendoza SN, Zhang L, Nielsen J, Lu H, Kerkhoven EJ. 2023. Yeast9: A Consensus Yeast Metabolic Model Enables Quantitative Analysis of Cellular Metabolism By Incorporating Big Data. *bioRxiv*. doi:10.1101/2023.12.03.569754

26th Feb 2025

Manuscript Number: MSB-2025-12861-T

Title: An enzyme activation network reveals extensive crosstalk between metabolic pathways

Dear Dr. Alam,

Please excuse my previous email in which we had an issue with our system and Reviewer 4 was listed incorrectly as Reviewer 1. I have corrected this in the reports below, all other contents of the email remain the same. Thank you for your understanding.

Thank you again for submitting your revised work to Molecular Systems Biology. We have now heard back from two of the original reviewers (Reviewers 2 and 3) who evaluated your study as well as a new reviewer (Reviewer 4) who replaced Reviewer 1 who was no longer able to re-review. As you will see below, the reviewers are supportive of the work but still have significant remaining concerns, and we would therefore ask you to address their concerns in a revision that will be re-reviewed. In particular it will be necessary to make the code and relevant data available in a public repository like Github (it is not sufficient to make these available upon request). In addition, some toning down of the conclusions and limitations will need to be further discussed in line with comments from Reviewers 2 and 3, and all issues on clarity raised by Reviewer 4 should be fully addressed. All other issues raised would need to be satisfactorily addressed. Please let me know in case you would like to discuss in further detail any of the comments, I would be happy to schedule a call.

We require:

1) A .docx formatted version of the manuscript text (including legends for main figures, EV figures and tables). Please make sure that the changes are highlighted to be clearly visible. Alternatively you may choose to submit your manuscript as a LaTeX file.

4) A .docx formatted letter INCLUDING the reviewers' reports and your detailed point-by-point responses to their comments. As part of the EMBO Press transparent editorial process, the point-by-point response is part of the Peer Review File (PRF), which will be published alongside your paper.

5) A complete author checklist, which you can download from our author guidelines (<https://www.embopress.org/page/journal/17574684/authorguide#submissionofrevisions>). Please insert information in the checklist that is also reflected in the manuscript. The completed author checklist will also be part of the PRF.

6) Please note that all corresponding authors are required to supply an ORCID ID for their name upon submission of a revised manuscript.

7) It is mandatory to include a 'Data Availability' section after the Materials and Methods. Before submitting your revision, primary datasets produced in this study need to be deposited in an appropriate public database, and the accession numbers and database listed under 'Data Availability'. Please remember to provide a reviewer password if the datasets are not yet public (see <https://www.embopress.org/page/journal/17574684/authorguide#dataavailability>).

In case you have no data that requires deposition in a public database, please state so in this section as follows: "This study includes no data deposited in external repositories". Note that the Data Availability Section is restricted to new primary data that are part of this study.

8) All Materials and Methods need to be described in the main text using our 'Structured Methods' format, which is required for all research articles. According to this format, the Methods section includes a Reagents and Tools Table (listing key reagents, experimental models, software and relevant equipment and including their sources and relevant identifiers) followed by a Methods and Protocols section describing the methods using a step-by-step protocol format. The aim is to facilitate adoption of the methodologies across labs. Please upload the Reagents and Tools table as a separate document when submitting your

revised manuscript. More information on how to adhere to this format as well as a downloadable template (.docx) for the Reagents and Tools Table can be found in our author guidelines:

<https://www.embopress.org/page/journal/17444292/authorguide#structuredmethods>

9) For data quantification: please specify the name of the statistical test used to generate error bars and p-values, the number (n) of independent experiments (specify technical or biological replicates) underlying each data point and the test used to calculate p-values in each figure legend. The figure legends should contain a basic description of n, p-values and the test applied. Graphs must include a description of the bars and the error bars (s.d., s.e.m.). Please provide exact p-values (in either the figure or figure legend).

10) Our journal encourages inclusion of *data citations in the reference list* to directly cite datasets that were re-used and obtained from public databases. Data citations in the article text are distinct from normal bibliographical citations and should directly link to the database records from which the data can be accessed. In the main text, data citations are formatted as follows: "Data ref: Smith et al, 2001" or "Data ref: NCBI Sequence Read Archive PRJNA342805, 2017". In the Reference list, data citations must be labeled with "[DATASET]". A data reference must provide the database name, accession number/identifiers and a resolvable link to the landing page from which the data can be accessed at the end of the reference. Further instructions are available at .

11) We replaced Supplementary Information with Expanded View (EV) Figures and Tables that are collapsible/expandable online. EV Figures should be cited as 'Figure EV1, Figure EV2' etc... in the text and their respective legends should be included in the main text after the legends of regular figures.

- Additional Tables/Datasets should be labeled and referred to as Table EV1, Dataset EV1, etc. Legends should be provided in a separate tab in case of .xls files. Alternatively, the legend can be supplied as a separate text file (README) and zipped together with the Table/Dataset file.

<https://www.embopress.org/page/journal/17574684/authorguide#expandedview>

12) Author contributions: CRedit has replaced the traditional author contributions section because it offers a systematic machine-readable author contributions format that allows for more effective research assessment. Please remove the Authors Contributions from the manuscript and use the free text boxes beneath each contributing author's name in our system to add specific details on the author's contribution. More information is available in our guide to authors.

13) Disclosure statement and competing interests: We updated our journal's competing interests policy in January 2022 and request authors to consider both actual and perceived competing interests. Please review the policy

<https://www.embopress.org/competing-interests> and update your competing interests if necessary.

14) Every published paper now includes a 'Synopsis' to further enhance discoverability. Synopses are displayed on the journal webpage and are freely accessible to all readers. They include a short stand first (maximum of 300 characters, including space) as well as 2-5 one-sentences bullet points that summarizes the paper. Please write the bullet points to summarize the key NEW findings. They should be designed to be complementary to the abstract - i.e. not repeat the same text. We encourage inclusion of key acronyms and quantitative information (maximum of 30 words / bullet point). Please use the passive voice. Please attach these in a separate file or send them by email, we will incorporate them accordingly.

Please note that these would be the final versions and changes during proofing are usually not allowed.

15) As part of the EMBO Publications transparent editorial process initiative (see our policy here:

https://www.embopress.org/transparent-process#Review_Process), Molecular Systems Biology will publish online a Peer Review File (PRF) to accompany accepted manuscripts.

In the event of acceptance, this file will be published in conjunction with your paper and will include the anonymous referee reports, your point-by-point response and all pertinent correspondence relating to the manuscript. Let us know whether you agree with the publication of the PRF and as here, if you want to remove or not any figures from it prior to publication.

Please note that the Author checklist will be published at the end of the PRF.

Molecular Systems Biology has a "scooping protection" policy, whereby similar findings that are published by others during review or revision are not a criterion for rejection. Should you decide to submit a revised version, I do ask that you get in touch after three months if you have not completed it, to update us on the status.

Yours sincerely,

Poonam Bheda, PhD
Scientific Editor
Molecular Systems Biology

Reviewer #2:

The authors have reasonably responded to the issues raised by myself and the other reviewers. They have repeated their study using a more recent yeast model, which resulted in similar findings as before. All reviewers commented on the issue use of using non-species-specific data from BRENDA. The authors discuss this limitation, but prefer to keep their methodology.

Although the paper has considerably improved in the last revision, I think there are still a few small issues that should be addressed.

- The differences in gene expression (between activated, non-activated, essential, non-essential enzymes) are statistically significant (due to sample size) but quite small in magnitude (Fig 3c). I would interpret this as a high confidence conclusion of there being very little difference. The differences between the medians (and between the standard deviations) are one order of magnitude smaller than the standard deviations themselves. I would be more conservative with the conclusions that are drawn from this.
- The authors say that they found activation of 140 enzymes in BRENDA, and this results in a final network with activation of 344 enzymes. How do the 140 later become 344?
- I agree with reviewer 1 that the discussion could go a bit deeper into the biological implications of the results, rather than just summarizing the main findings.
- For instance, the authors analysed gene co-expression patterns between the enzymes that produce an activator compound and the enzymes regulated by that compound. With this information, the authors create a sub-network of the original network. There is not much for the reader to learn from this subnetwork. Why would the cell co-regulate the gene expression of the activatory compound and the respective enzyme, in addition to the allosteric regulation? These kind of questions deserve a deeper discussion.
- In response to my comment about sharing the code and data in a github repo, the authors say that the code and data are made available upon request. I think "available upon request" is not an acceptable data availability statement. I don't see any reasonable justification for a paper in 2025 not to share all the data and code.

Reviewer #3:

The article by Al Zubaidi et al integrates enzyme activation networks to assess regulatory crosstalk between pathways. I have previously reviewed this article and find it to be of exceptional quality. One issue that I would like to raise is that much of the data reported in BRENDA is not fully reflective of the in vivo situation with much being collected at optimal in vitro conditions for the enzymes as opposed to the cellular pH, ion concentrations etc. I think that the authors need to caveat their findings with a comment to this effect or even better assess the possibility of this effecting the outcome of their results. Otherwise I would like to commend the authors on an excellent piece of work.

Reviewer #4:

Title: An enzyme activation network provides evidence for extensive regulatory crosstalk between metabolic pathways.
Authors: Sultana Mohammed Al Zubaidi, Muhammad Ibtisam Nasar, Richard A. Notebaart, Markus Ralser, Mohammad Tauqeer Alam
Journal: Molecular Systems Biology

Reviewer recommendation: Acceptable with major revisions.

Reviewer summary:

Collected activation information from BRENDA

Overlaid it on the Yeast9 model

Did a pathway-level analysis of activation to determine how much metabolic pathways are controlled by activation, and from where.

Key biomass pathways (nucleotides, amino acids, glycolysis, and pyruvate metabolism) have high levels of enzyme activation.

Positive regulation early in pathways (first three enzymatic steps).

No pattern in activation interactions and essentiality.

Found that activator metabolites are closer in the metabolic network to glucose than non-activators.

Summary of reviewer concerns:

Overall, the research is of interest to the community. However, there are a number of instances of poor communication of results that make it difficult to evaluate the merits of this manuscript. This makes the manuscript appear to be an advanced draft, rather than something ready for publication. A key concern when evaluating this manuscript is that number do not seem to line up in the reported results with data sources, or contradictory statements about number are made, see minor concerns 2, 3, and 4. Due to such contradictory statements, it is hard to evaluate the

Major Concerns:

1. Code availability: Code should be made available through a public repository, otherwise analysis techniques may be lost if the corresponding author were to become unreachable.
2. There are too many issues of lack of clarity in this manuscript. Individual instances are described under "minor concerns" and "issues of spelling, grammar, and clarity". However, the number of these instances make this theme rise to a major concern.

Minor Concerns:

1. Page 2, lines 14 to 15: Missing uncompetitive regulation, which is something most students are introduced to along with Michaelis-Menten kinetics.
2. Page 3, lines 38 to 39: The numbers presented for enzyme activation do not seem to line up. In page 3 line 27, it is noted that the BRENDA contains evidence of activation interactions for 140 enzymes. However, Page 3, lines 38 to 39 state that the final regulatory network developed contains 1499 total interactions for 344 enzymes. The paragraph starting from page 3 line 31 is written such that the reader assumes that the network flows from the BRENDA evidence. This is further reinforced by figure 1a. Therefore, where does the evidence for metabolic interactions for the additional 204 enzymes come from? It is not clear in section.
3. Page 4, lines 21 to 32: Similarly, it is claimed that 54% of enzymes in the Yeast9 model have activation interactions. However, later in this paragraph it is stated that no individual category of enzyme has more than 40% of its members having activation reactions. As these end up effectively averaged over the whole model, what pathway have more than half of their members with an activation interaction? Indeed, in figure 1f it is shown that no class of enzyme has more than 35% activated and non-activated, so how does the total rise to 54%?
4. Page 5, lines 26 to 37: Again, there is an issue with clarity of numbers here. It is stated (line 27) that 36 pathways have significant activation interactions with other pathways. Yet it says in line 31 that 26 of these 36 either activate or are activated by other pathways. These statements are mutually exclusive, where line 27 states that 36 pathways have activation interactions with other pathways, and line 31 states that this is only the case for 26 pathways. Which is it? In the case of line 31, what does it mean to "who significant activation interactions with other pathways" yet not "either activate or are activated by other pathways"?
5. Page 6, line 19: How is a metabolite removed from a model? Is it that all reactions which use that metabolite are blocked?
6. Figure 1b: grey edges should not be used in combination with grey nodes, as it makes the nodes difficult to see closer to the center of the network. This is particularly important as the manuscript notes that many of the most-connected enzymes are non-essential (hence colored grey with many grey lines).
7. Figure 2a: What does it mean to be a reaction which has activation, but is labeled as "not activated"?
8. Figure 4a: What is meant by the "shortest path length between nutrient and other metabolites"? Is this similar to the shortest path length from glucose, but for any nutrient (e.g. nitrogen, sulfur, and phosphorus sources)? Or is this for only carbon nutrients. If so, which? Note that Page 7 lines 40 to 44 seem to indicate that this is the distance from glucose only.
9. Page 7, lines 1 to 5: These analyses require statistical validation, like other assertions of significance made throughout this manuscript. Figure 3 c iv appears to show almost no significant differences between the four conditions shown when quartile ranges are taken into account, and a statistical test would be needed to convince most readers that differences exist. Discussion following these statements are moot due to lack of convincing difference.
10. Figure 3 c iv: The box-and-whisker plots for both non-activated categories lack a marker for the location of the median.
11. Page 7, line 41: Is this a test of "mode" then? If so, state that this is the statistical test used.
12. Supplementary figures are reproduced in the in-manuscript figures. For example, Supplementary Figure S4 is identical to Figure 4 c iv.
13. Number of supplementary figures: there are more supplementary figures provided (5 in total) than are given in the supplementary information on page 14, lines 30 to 42

Concerns of spelling, grammar, and clarity:

1. Page 2, lines 4 and 5: If regulatory interactions are occurring at different levels of the cellular process, then they are different types of interactions. Therefore, its unclear how the label "most common" applies.
2. Page 3, lines 31 to 36: It is unclear what is meant by "non-cellular molecules". Are these molecules that do not generally exist within the cell (such as drugs, anti-fungal compounds, etc.)?
3. Page 4, line 8: KEGG is an acronym (like BRENDA) and should be capitalized.
4. Page 6, line 10: the prefix "non" is usually followed by a dash, like "non-essential". This happens in several other places throughout the manuscript.
5. Reporting statistical results. Statistical results have formatting inconsistencies in reporting. In some cases, the test is listed first, then the p-value cut off used for significance. In others, the p-value of the statistic is reported first, followed by the test used.
6. Figures 1 e and f: first letters of titles should be capitalized.
7. Figure 1b: scientific names should be italicized.
8. Overall for pie-charts: Please label pie charts with percentages or numbers throughout (such as done in Figures 1 c and d, but not in figures 2 a and b).
9. Figure 4a: Spelling mistake in y-axis label of "metablites".
10. Figures: Are the figures provided at the end of the manuscript figures that will be included within the body of the manuscript (as indicated by their figure captions given as "Figure 1", "Figure 2", etc. on page 16) or are they supplemental figures (for instance, on page 7 line 2, Supplementary Figure S3a is reference, which appears to be the same as Figure 3a).
11. Figure complexity: Some graphs should become their own sub-figure, rather than remain as sub-sub-figures (example: Figures 3 c i to iv should become Figures 3 d to g).

Rev_Com_number: RC-2023-02262

New_manu_number: MSB-2025-12861-T

Corr_author: Alam

Title: An enzyme activation network reveals extensive crosstalk between metabolic pathways

**Reviewer #2:**

**The authors have reasonably responded to the issues raised by myself and the**
**other reviewers. They have repeated their study using a more recent yeast model,**
**which resulted in similar findings as before. All reviewers commented on the**
**issue use of using non-species-specific data from BRENDA. The authors discuss**
**this limitation, but prefer to keep their methodology.**

**Although the paper has considerably improved in the last revision, I think there**
**are still a few small issues that should be addressed.**

**Response:** We sincerely appreciate the reviewer's positive assessment of our work.
We have incorporated their suggestions into the manuscript and addressed their
concerns below.

**- The differences in gene expression (between activated, non-activated, essential,**
**non-essential enzymes) are statistically significant (due to sample size) but quite**
**small in magnitude (Fig 3c). I would interpret this as a high confidence**
**conclusion of there being very little difference. The differences between the**
**medians (and between the standard deviations) are one order of magnitude**
**smaller than the standard deviations themselves. I would be more conservative**
**with the conclusions that are drawn from this.**

**Response:** We acknowledge the Reviewer's concern regarding the relatively small
magnitude of gene expression differences across these categories.
The p-value may be significant on the basis of a large sample size, arguably, the effect
size remains low, which makes the biological relevance hard to interpret. However, this
low effect size stems from a highly heterogeneous dataset summarizing enzymes of
very different function, that is also the source in the large SDs. To avoid
overinterpretation, we have re-written this section carefully. Furthermore, we conducted
additional analyses by calculating Cohen's d, a standardized effect size that measures
the difference between two group means in terms of standard deviation units. For
essential vs. non-essential genes, Cohen's d is 0.3, indicating a small but meaningful
difference. In contrast, the effect size for activated vs. non-activated enzymes is 0.12,
suggesting a minimal difference with limited biological impact. However, when
comparing enzymes that are both essential for growth and intracellularly activated
against enzymes that are non-essential for growth and non-activated, we observe a
more substantial difference (Cohen's d = 0.4), suggesting a meaningful distinction
between these groups. These findings have been added in the manuscript.

*It has previously been reported that essential enzymes are tightly regulated to maintain constant*
*gene expression levels regardless of genetic or environmental perturbations (Grüning et al.,*
*2010; Yang et al., 2012). Consequently, these genes are expected to exhibit less variation in gene*

*expression (Park and Lehner, 2013). Consistent with those studies (Grüning et al., 2010; Park*
*and Lehner, 2013; Yang et al., 2012), using microarray gene expression data across hundreds of*
*conditions, we find that essential enzymes (whether activated or non-activated) have higher gene*
*expression levels compared to non-essential enzymes, and effect size is small but having*
*meaningful difference (median log₂ gene expression 10.78 vs. 10.11, SD 1.66 vs 1.92, Cohen's d*
*= 0.303, p-value < 2.2e-16 T-test, Supplementary Figure S3a). We did not observe any*
*substantial and meaningful difference in the gene expression levels between activated and non-*
*activated enzymes, regardless of their essentiality (median log₂ gene expression 10.47 vs. 10.22,*
*SD 1.84 vs 1.85, Cohen's d = 0.12, p-value < 2.2e-16 T-test, Supplementary Figure S3b).*
*However, enzymes that are essential for growth and are intracellularly activated exhibit the*
*highest median gene expression values and the least variance in contrast to enzymes that are*
*non-essential for growth and rely on activation for their expression (median log₂ gene*
*expression 10.87 vs 10.0, SD 1.67 vs 1.92, Cohen's d = 0.4 between the two groups, p-value <*
*2.2e-16 T-test, Figure 3c iv). This indicates that these essential activated enzymes are*
*consistently upregulated and maintain the highest expression levels in comparison with non-*
*essential non-activated enzymes. The Cohen's d that measure of effect size between essential plus*
*activated and non-essential plus non-activated is 0.4, suggesting a meaningful difference*
*between these groups.*

**- The authors say that they found activation of 140 enzymes in BRENDA, and this**
**results in a final network with activation of 344 enzymes. How do the 140 later**
**become 344?**

**Response:** We apologize for any confusion and lack of clarity in the manuscript.
Reviewer #3 has raised a similar concern, and we have provided a detailed response
on page 5 line 20 onwards. In summary, the BRENDA database contains kinetic
information for 140 *S. cerevisiae* enzymes, of which 63% share at least one activation
interaction with the same activator molecule across different species. By incorporating
enzyme kinetic data from multiple species, we were able to construct the activation
network with 344 enzymes.

**- I agree with reviewer 1 that the discussion could go a bit deeper into the**
**biological implications of the results, rather than just summarizing the main**
**findings.**

**- For instance, the authors analysed gene co-expression patterns between the**
**enzymes that produce an activator compound and the enzymes regulated by that**
**compound. With this information, the authors create a sub-network of the original**
**network. There is not much for the reader to learn from this subnetwork. Why**
**would the cell co-regulate the gene expression of the activatory compound and**

**the respective enzyme, in addition to the allosteric regulation? These kind of**
**questions deserve a deeper discussion.**

**Response:** We have expanded the discussion and included the below section.

*Moreover, in nearly one-third of the activatory interactions, a gene-producing activator is co-*
*expressed with activated enzymes. This may be associated with positive feedback loops or co-*
*regulated gene networks, where the activator enhances the expression of enzymes involved in a*
*biochemical pathway. For example, transcriptional activators can be co-expressed with the*
*enzymes they regulate. Similarly, in certain signaling cascades, an activator protein may*
*regulate its own expression. Additionally, we observe that in one-third of the activatory*
*interactions, a gene-producing activator also engages in genetic interactions.*

**- In response to my comment about sharing the code and data in a github repo,**
**the authors say that the code and data are made available upon request. I think**
**"available upon request" is not an acceptable data availability statement. I don't**
**see any reasonable justification for a paper in 2025 not to share all the data and**
**code.**

**Response:** All the codes and relevant data is available through github. Please see the
link here – <https://github.com/mdtauqeer/activationNetwork>

**Reviewer #3:**

**The article by Al Zubaidi et al integrates enzyme activation networks to assess**
**regulatory crosstalk between pathways. I have previously reviewed this article**
**and find it to be of exceptional quality. One issue that I would like to raise is that**
**much of the data reported in BRENDA is not fully reflective of the in vivo situation**
**with much being collected at optimal in vitro conditions for the enzymes as**
**opposed to the cellular pH, ion concentrations etc. I think that the authors need to**
**caveat their findings with a comment to this effect or even better assess the**
**possibility of this effecting the outcome of their results. Otherwise I would like to**
**commend the authors on an excellent piece of work.**

**Response:** We totally agree, this remains a limitation not only of our paper but of most
of today's enzymology. We have highlighted this in the discussion. We hope that our
paper further stimulates the field to develop strategies to better study these processes
in the future in vivo.

**Reviewer #4:**

**Title: An enzyme activation network provides evidence for extensive regulatory**
**crosstalk between metabolic pathways.**

**Authors: Sultana Mohammed Al Zubaidi, Muhammad Ibtisam Nasar, Richard A.**
**Notebaart, Markus Ralser, Mohammad Tauqeer Alam**

**Journal: Molecular Systems Biology**

**Reviewer recommendation: Acceptable with major revisions.**

**Reviewer summary:**

**Collected activation information from BRENDA**

**Overlaid it on the Yeast9 model**

**Did a pathway-level analysis of activation to determine how much metabolic**
**pathways are controlled by activation, and from where.**

**Key biomass pathways (nucleotides, amino acids, glycolysis, and pyruvate**
**metabolism) have high levels of enzyme activation.**

**Positive regulation early in pathways (first three enzymatic steps).**

**No pattern in activation interactions and essentiality.**

**Found that activator metabolites are closer in the metabolic network to glucose**
**than non-activators.**

**Summary of reviewer concerns:**

**Overall, the research is of interest to the community. However, there are a**
**number of instances of poor communication of results that make it difficult to**
**evaluate the merits of this manuscript. This makes the manuscript appear to be**
**an advanced draft, rather than something ready for publication. A key concern**
**when evaluating this manuscript is that number do not seem to line up in the**
**reported results with data sources, or contradictory statements about number are**
**made, see minor concerns 2, 3, and 4. Due to such contradictory statements, it is**
**hard to evaluate the**

**Response:** We sincerely appreciate the reviewer's positive overall assessment of our
work. We apologize for any language inconsistencies in the paper that may have given
the impression of a draft-stage submission. We have thoroughly revised the manuscript
to address the reviewer's concerns and enhance its clarity and readability. Additionally,
we have provided detailed clarifications regarding the numbers put at question, as
outlined in our point-by-point response below. We hope these revisions resolve any
misunderstandings.

**Major Concerns:**

**1. Code availability: Code should be made available through a public repository,**
**otherwise analysis techniques may be lost if the corresponding author were to**
**become unreachable.**

**Response:** The Reviewer #2 has also asked for the codes to be available. We confirm
the codes, and the model produced in this study are made available through the github
repository. Please see the link here – <https://github.com/mdtauqeer/activationNetwork>

**2. There are too many issues of lack of clarity in this manuscript. Individual**
**instances are described under "minor concerns" and "issues of spelling,**
**grammar, and clarity". However, the number of these instances make this theme**
**rise to a major concern.**

**Response:** We truly appreciate the reviewer's careful review, which helped identify
these overlooked issues. Their attention to detail has been invaluable in improving our
manuscript, and we are grateful for their insightful feedback.

**Minor Concerns:**

**1. Page 2, lines 14 to 15: Missing uncompetitive regulation, which is something**
**most students are introduced to along with Michaelis-Menten kinetics.**

**Response:** The uncompetitive regulation type has been mentioned in the main text.

**2. Page 3, lines 38 to 39: The numbers presented for enzyme activation do not**
**seem to line up. In page 3 line 27, it is noted that the BRENDA contains evidence**
**of activation interactions for 140 enzymes. However, Page 3, lines 38 to 39 state**
**that the final regulatory network developed contains 1499 total interactions for**
**344 enzymes. The paragraph starting from page 3 line 31 is written such that the**
**reader assumes that the network flows from the BRENDA evidence. This is**
**further reinforced by figure 1a. Therefore, where does the evidence for metabolic**
**interactions for the additional 204 enzymes come from? It is not clear in section.**

**Response:** We thank the reviewer for pointing this out. Reviewer #1 also raised a
similar concern. We agree that the original statement lacked clarity, and we have
revised it accordingly. As mentioned in the manuscript, the Yeast9 model served as the
foundation for constructing the activation network, and we incorporated cross-species
enzyme kinetic data to develop it. Specifically, we analyzed 635 metabolic enzymes
from the Yeast9 model to compile a list of associated activatory molecules across
species. To justify the use of cross-species data, we examined how many of these 635
*S. cerevisiae* enzymes had activation evidence from both *S. cerevisiae* and other
species in the BRENDA database. Among them, we identified kinetic information for
140 *S. cerevisiae*-specific enzymes. Notably, 88 of these enzymes (63%) shared at
least one activation interaction with the same activator molecule across different
species. This high level of concordance supports our assumption that activation

interactions are conserved across species. Subsequently, we mapped the cross-
species enzyme kinetics data onto the Yeast9 model and excluded all non-cellular
metabolites. The final activation network consists of 1,499 activatory interactions
involving 344 enzymes and 286 cellular metabolites. These enzymes and metabolites
from Yeast9 model however the activation interaction evidence is from cross species.
We have now revised the text to avoid any confusion.:

*To construct a genome-scale enzyme-metabolite activation interactions, we obtained the*
*topology of the Saccharomyces cerevisiae metabolic network and enriched it with cross-species*
*acquired data of enzyme activation. For each metabolic enzyme (in total 635 enzymes) in the*
*genome-scale metabolic model (Yeast9) (Zhang et al., 2023) a list of all associated activatory*
*molecules were downloaded from the Brenda database, using SOAP clients instructions (Chang*
*et al., 2021). To justify the use of cross-species data, we examined how many of these 635 S.*
*cerevisiae enzymes had activation evidence from both S. cerevisiae and other species in the*
*BRENDA database. Among them, we identified kinetic information for 140 S. cerevisiae-specific*
*enzymes. Notably, 88 of these enzymes (63%) shared at least one activation interaction with the*
*same activator molecule across different species. This high level of concordance supports our*
*assumption that activation interactions are conserved across species.*

*The obtained list of activators includes various molecules, ranging from non-cellular compounds*
*like drugs to intracellular metabolites produced within the cell. Enzyme assays confirm that all*
*these metabolites enhance the enzyme's activity. By comparing the list of activatory molecules*
*with the intracellular metabolites of the metabolic model (Zhang et al., 2023) all non-cellular*
*molecules were removed (Figure 1a; see the method section for details) to produce the cell-*
*intrinsic activation interactions network (Figure 1b, Supplementary File 1). In the network, the*
*nodes represent enzymes and activator metabolites, and the edges are formed between nodes*
*when an enzyme is activated by a metabolite (Figure 1b). The final network comprises 1,499*
*activatory interactions involving 344 enzymes and 286 cellular metabolites. While these enzymes*
*and metabolites are derived from the Yeast9 model, the activation interaction evidence is based*
*on cross-species data.*

**3. Page 4, lines 21 to 32: Similarly, it is claimed that 54% of enzymes in the Yeast9**
**model have activation interactions. However, later this this paragraph it is stated**
**that no individual category of enzyme has more than 40% of its members having**
**activation reactions. As these end up effectively averaged over the whole model,**
**what pathway have more than half of their members with an activation**
**interaction? Indeed, in figure 1f it is shown that no class of enzyme has more**
**than 35% activated and non-activated, so how does the total rise to 54%?**

**Response:** Again apologies if our description created the confusion. We have carefully
reviewed the numbers given, but they are correct. On page 3 and 4, we state that 54%
of *S. cerevisiae* enzymes (that is 344 out of the total 635 enzymes) have intracellular
activation interactions. We are examining the distribution of these 54% intracellular
activated enzymes in various enzyme classes. So, the percentages in the later section
are calculated from the intracellular activated enzymes (344). We have added a better
clarification to the main text.

**4. Page 5, lines 26 to 37: Again, there is an issue with clarity of numbers here. It is**
**stated (line 27) that 36 pathways have significant activation interactions with**
**other pathways. Yet it says in line 31 that 26 of these 36 either activate or are**
**activated by other pathways. These statements are mutually exclusive, where line**
**27 states that 36 pathways have activation interactions with other pathways, and**
**line 31 states that this is only the case for 26 pathways. Which is it? In the case of**
**line 31, what does it mean to "who significant activation interactions with other**
**pathways" yet not "either activate or are activated by other pathways"?**

**Response:** We have revised the text to provide a clearer explanation of the results. We
hope this resolves any confusion.

*Among the 72 metabolic pathways in the Yeast9 network, half (36, or 50%) exhibit significant*
*activation interactions with other pathways (p -value < 0.05 , hypergeometric test). These*
*pathways either activate, are activated by, or both activate and are activated by other pathways.*
*The remaining pathways, despite containing activated enzymes, do not show significant*
*activation interactions with other pathways (p -value < 0.05 , hypergeometric test). Among the 36*
*pathways with activation interactions, the majority (26, or 72%) engage in only one type of*
*interaction—either being activated by other pathways without activating any or activating other*
*pathways without being activated themselves. Additionally, we observe that 13 pathways activate*
*only a single pathway, 17 are activated by just one pathway, and only 4 pathways exhibit*
*significant self-activation. This pattern highlights the general specificity of activation*
*interactions within the network.*

**5. Page 6, line 19: How is a metabolite removed from a model? Is it that all**
**reactions which use that metabolite are blocked?**

**Response:** The reviewer is correct. To remove a metabolite from the model, we block
all associated reactions by setting both the lower and upper flux bounds to zero. This
has been mentioned in the method section as follows.

*For an enzyme (or metabolite) knockout, both the lower and upper flux bounds of all*
*associated reactions were set to zero.*

**6. Figure 1b: grey edges should not be used in combination with grey nodes, as it**
**makes the nodes difficult to see closer to the center of the network. This is**
**particularly important as the manuscript notes that many of the most-connected**
**enzymes are non-essential (hence colored grey with many grey lines).**

**Response:** The edge color has been changed to light blue, making it clearly
distinguishable.

**7. Figure 2a: What does it mean to be a reaction which has activation, but is**
**labeled as "not activated"?**

**Response:** Figure 2a is showing a pie chart which shows the number of pathways
associated with intracellularly activated (65/67), extracellularly activated (1/67), and
non-activated (1/67). There is no reaction which has activation but labelled as not-
activated. We are sorry for this confusion.

**8. Figure 4a: What is meant by the "shortest path length between nutrient and**
**other metabolites"? Is this similar to the shortest path length from glucose, but for**
**any nutrient (e.g. nitrogen, sulfur, and phosphorus sources)? Or is this for only**
**carbon nutrients. If so, which? Note that Page 7 lines 40 to 44 seem to indicate**
**that this is the distance from glucose only.**

**Response:** The reviewer is correct. Nutrients represent all exchange metabolites of the
Yeast9 model, including carbon, nitrogen, sulfur, and phosphorus sources. The shortest
path length between nutrients and other metabolites in the manuscript refers to the
shortest path length from a given metabolite to all exchange metabolites. In the
manuscript, Figure 4c specifically represents the shortest path length from glucose.
However, Supplementary Figure S5 presents the results for the shortest path length
from all exchange metabolites. We have added a clarification to the text.

**9. Page 7, lines 1 to 5: These analyses require statistical validation, like other**
**assertions of significance made throughout this manuscript. Figure 3 c iv**
**appears to show almost no significant differences between the four conditions**
**shown when quartile ranges are taken into account, and a statistical test would**
**be needed to convince most readers that differences exist. Discussion following**
**these statements are moot due to lack of convincing difference.**

**Response:** We performed additional analyses using Cohen's d, a standardized effect
size that quantifies the difference between two group means in terms of standard

deviation units. Initially, we used a t-test to assess these comparisons, which yielded
extremely small p-values. However, as Reviewer #2 pointed out, this could be attributed
to the large sample size rather than a meaningful effect. From this analysis now report
Cohen's d as a more informative measure of effect size. Please see our response on
Page 1-2 for more details.

**10. Figure 3 c iv: The box-and-whisker plots for both non-activated categories**
**lack a marker for the location of the median.**

**Response: This has been corrected, thank you.**

**11. Page 7, line 41: Is this a test of "mode" then? If so, state that this is the**
**statistical test used.**

**Response:** Unfortunately, we are unsure of the reviewer's request. We used a t-test in
the analysis presented on page 7, line 41, and this is explicitly mentioned in the text.

**12. Supplementary figures are reproduced in the in-manuscript figures. For**
**example, Supplementary Figure S4 is identical to Figure 4 c iv.**

**Response:** We believe the reviewer is referring to Figure 3C iv and Supplementary
Figure S4. These figures present different data: Figure 3C iv depicts gene expression
values, while Supplementary Figure S4 shows protein expression levels. The latter was
generated in response to the previous review. However, we acknowledge the reviewer's
observation that both figures appear similar and lead to the same conclusion, which was
a key point of the analysis.

**13. Number of supplementary figures: there are more supplementary figures**
**provided (5 in total) than are given in the supplementary information on page 14,**
**lines 30 to 42**

**Response:** We apologize for this oversight and appreciate the reviewer for bringing it to
our attention. While all supplementary figures are cited in the main text, some were
mistakenly omitted from the list of Supplementary Information. We have now corrected
this error.

**Concerns of spelling, grammar, and clarity:**

**1. Page 2, lines 4 and 5: If regulatory interactions are occurring at different levels**
**of the cellular process, then they are different types of interactions. Therefore, its**
**unclear how the label "most common" applies.**

**Response:** We apologise for the typo. The statement has been corrected as below.

*To survive in changing environments, the regulation of cellular metabolism through*
*tuning enzyme abundance or activities is essential (Klosik et al., 2017). Cells achieve*
*metabolic regulation by a range of regulatory interactions (Metallo and Vander Heiden,*
*2013; Zhu and Thompson, 2019), which occur at different levels of cellular processes.*

**2. Page 3, lines 31 to 36: It is unclear what is meant by "non-cellular molecules".**
**Are these molecules that do not generally exist within the cell (such as drugs,**
**anti-fungal compounds, etc.)?**

**Response:** The reviewer is correct. By "non-cellular molecules," we refer to drugs that
are not produced within the cell. We have revised the main text to clarify this.

*The obtained list of activators includes various molecules, ranging from non-cellular*
*compounds like drugs to intracellular metabolites produced within the cell. Enzyme*
*assays confirm that all these metabolites enhance the enzyme's activity.*

**3. Page 4, line 8: KEGG is an acronym (like BRENDA) and should be capitalized.**

**Response:** Corrected, thank you!

**4. Page 6, line 10: the prefix "non" is usually followed by a dash, like "non-**
**essential". This happens in several other places throughout the manuscript.**

**Response:** Corrected, thank you!

**5. Reporting statistical results. Statistical results have formatting inconsistencies**
**in reporting. In some cases, the test is listed first, then the p-value cut off used**
**for significance. In others, the p-value of the statistic is reported first, followed by**
**the test used.**

**Response:** we would like to thank the reviewer for notifying this. We have corrected this
issue throughout the manuscript.

**6. Figures 1 e and f: first letters of titles should be capitalized.**

**Response:** Corrected, thank you!

**7. Figure 1b: scientific names should be italicized.**

Response: Corrected, thank you!

8. Overall for pie-charts: Please label pie charts with percentages or numbers throughout (such as done in Figures 1 c and d, but not in figures 2 a and b).

Response: We have added the numbers in the pie chart.

9. Figure 4a: Spelling mistake in y-axis label of "metablites".

Response: Corrected. Thank you!

10. Figures: Are the figures provided at the end of the manuscript figures that will be included within the body of the manuscript (as indicated by their figure captions given as "Figure 1", "Figure 2", etc. on page 16) or are they supplemental figures (for instance, on page 7 line 2, Supplementary Figure S3a is reference, which appears to be the same as Figure 3a).

Response: Figures such as Figure 1 and Figure 2 are the main figures included within the manuscript, while supplementary figures are labeled as Supplementary Figure S1, Supplementary Figure S2, etc., and are provided in the supplementary material. Supplementary Figure S3a and Figure 3a are two different figures. Figure 3a presents the percentage prevalence of essential, non-essential, activated enzyme (or activator metabolites), and non-activated enzyme (or non-activator metabolites). In contrast, Supplementary Figure S3a illustrates differences in gene expression profiles for essential and non-essential enzymes, regardless of whether they are intracellularly activated.

11. Figure complexity: Some graphs should become their own sub-figure, rather than remain as sub-sub-figures (example: Figures 3 c i to iv should become Figures 3 d to g).

Response: The reason why we choose this figure layout is as follows: All subfigures of Figure 3c illustrate the degree distribution of enzymes and the variation in gene expression across different enzyme categories. Subfigures ii, iii, and iv of Figure 3c are derived from Figure 3c i, summarizing the data in a more simplified manner. Similarly, the subfigures of Figure 3b represent the degree distribution of activators, with Figures 3b ii and iii being derived from Figure 3b i. We believe this structure makes an ideal compromise between accurate data representation and making it easy to read.

10th Apr 2025

Manuscript Number: MSB-2025-12861R

Title: An enzyme activation network reveals extensive crosstalk between metabolic pathways

Dear Dr. Alam,

Thank you for the submission of your revised manuscript to Molecular Systems Biology. I am pleased to inform you that we will be able to accept your manuscript pending the following final amendments:

- 1) Please check the "Author Checklist" carefully and complete all relevant questions. Responses in Column D need to be selected from the drop-down menu, and the responses in Column E should be completed only for positive responses.
- 2) In the main manuscript file, please include keywords to max. 5.
- 3) Please format the Data availability section according to the example below:
"The datasets and computer code produced in this study are available in the following databases:
- Chip-Seq data: Gene Expression Omnibus GSE46748 (<https://www.ncbi.nlm.nih.gov/geo/query/acc.cgi?acc=GSE46748>)
- Modeling computer scripts: GitHub (<https://github.com/SysBioChalmers/GECKO/releases/tag/v1.0>)
- [data type]: [full name of the resource] [accession number/identifier] ([doi or URL or identifiers.org/DATABASE:ACCESSION])"
- 4) Code: The link for the codes for data analysis available in Github in the Data Availability section does not work. Please update the link to direct to the proper website. Please also be sure that a README file with practical use instructions for potential future users of your code is included in Github.
- 5) Please rename "Competing Interest" to "Disclosure and competing interests statement". We updated our journal's competing interests policy in January 2022 and request authors to consider both actual and perceived competing interests. Please review the policy <https://www.embopress.org/competing-interests> and update your competing interests if necessary.
- 6) Our journal encourages inclusion of *data citations in the reference list* to directly cite datasets that were re-used and obtained from public databases. Data citations in the article text are distinct from normal bibliographical citations and should directly link to the database records from which the data can be accessed. In the main text, data citations are formatted as follows: "Data ref: Smith et al, 2001" or "Data ref: NCBI Sequence Read Archive PRJNA342805, 2017". In the Reference list, data citations must be labeled with "[DATASET]". A data reference must provide the database name, accession number/identifiers and a resolvable link to the landing page from which the data can be accessed at the end of the reference. Further instructions are available at .
- 7) The "Methodology" section should be renamed to "Methods".
- 8) All Materials and Methods need to be described in the main text using our 'Structured Methods' format. According to this format, the Methods section includes a Reagents and Tools Table (listing key reagents, experimental models, software and relevant equipment and including their sources and relevant identifiers) followed by a Methods and Protocols section describing the methods, ideally using a step-by-step protocol format. The aim is to facilitate adoption of the methodologies across labs. Please download and fill our Reagents and Tools Table template (.docx), which you can find in our author guidelines: <https://www.embopress.org/page/journal/14693178/authorguide#structuredmethods>.
When submitting your revised manuscript, please do not include the Reagents and Tools Table in the Methods section of the manuscript but upload it as a separate file choosing the file type "Reagent Table".
An example of a Method paper with Structured Methods can be found here: <https://www.embopress.org/doi/10.15252/msb.20178071> . "
- 9) Please place individual sections of the manuscript in the following order: Title page - Abstract & Keywords - Introduction - Results - Discussion - Methods - Data Availability - Acknowledgements - Disclosure and Competing Interests Statement - References - Figure Legends - Expanded View Figure Legends.
- 10) For the figures and figure legends, please take care of the following:
 - Please upload Figures EV1-5 as individual Figure files (like the main figures) with legends included in main manuscript file.
 - Please make sure to update the callouts of all figures in the main manuscript text. Currently panel callouts are missing for Figure EV2 and Figure EV3.
 - Please indicate the statistical test used for data analysis in the legends of figures 3B, 4D
 - Please note that the box plots need to be defined in terms of minima, maxima, centre, bounds of box and whiskers, and percentile in the legends of figures 3B, 4A, B; EV3 A-C; EV4
 - Please note that information related to n is missing in the legends of figures 3B, 4A, B; EV3 A-C; EV5
- 11) Dataset EV legend: The source file name, title and legend for Supplementary File 1 all need to be updated to Dataset EV1. This file should be uploaded individually as a Dataset file with the legend removed from the main manuscript file and included as a separate tab/sheet in the Excel file. Please also ensure that the callouts for this file are correct in the main manuscript.
- 12) Appendix file: Please upload the Appendix as a single PDF (no separate image files are needed).
- 13) Synopsis:
 - Synopsis image: Please provide a graphic that summarises the main findings of the manuscript on a glance and upload it as a high-resolution jpeg file 550 pixels wide x (300-600) pixels high.
 - Synopsis text: Please provide a short standfirst (maximum of 300 characters, including space), limit the bullet points to max. 5 and upload it as a separate .doc file. Please write the bullet points to summarise the key NEW findings. They should be

designed to be complementary to the abstract - i.e. not repeat the same text. We encourage inclusion of key acronyms and quantitative information (maximum of 30 words / bullet point). Please use the passive voice.

14) As part of the EMBO Publications transparent editorial process initiative (see our policy here:

https://www.embopress.org/transparent-process#Review_Process), Molecular Systems Biology will publish online a Peer Review File (PRF) to accompany accepted manuscripts. This file will be published in conjunction with your paper and will include the anonymous referee reports, your point-by-point response and all pertinent correspondence relating to the manuscript. Let us know whether you agree with the publication of the PRF and as here, if you want to remove or not any figures from it prior to publication. Please note that the Authors checklist will be published at the end of the PRF.

15) After your paper is published, we will promote it on social media. If you have any handles or hashtags for Bluesky you would like included, please let us know.

16) Please provide a point-by-point letter INCLUDING my comments and your detailed responses (as Word file).

I look forward to reading a new revised version of your manuscript as soon as possible.

Yours sincerely,

Poonam Bheda, PhD
Scientific Editor
Molecular Systems Biology

Reviewer #1:

Overall, the authors did a good job of addressing the issues of clarity noted by the reviewers. The current manuscript is much improved and the strength of the work done is clear. As is, this manuscript is suitable for publication.

Reviewer #2:

The authors have properly addressed all my comments. I congratulate them on a very solid piece of work.

Thank you for the submission of your revised manuscript to Molecular Systems Biology. I am pleased to inform you that we will be able to accept your manuscript pending the following final amendments:

Response: Thank you very much for this wonderful news. I am delighted to read your email and learn about the conditional acceptance of our manuscript. Below, I provide a point-by-point response to all your comments and suggestions.

1) Please check the "Author Checklist" carefully and complete all relevant questions. Responses in Column D need to be selected from the drop-down menu, and the responses in Column E should be completed only for positive responses.

Response: Author Checklist is revised.

2) In the main manuscript file, please include keywords to max. 5.

Response: 5 keywords are added.

3) Please format the Data availability section according to the example below:
"The datasets and computer code produced in this study are available in the following databases:

- Chip-Seq data: Gene Expression Omnibus GSE46748

(<https://www.ncbi.nlm.nih.gov/geo/query/acc.cgi?acc=GSE46748>)

- Modeling computer scripts: GitHub

(<https://github.com/SysBioChalmers/GECKO/releases/tag/v1.0>)

- [data type]: [full name of the resource] [accession number/identifier] ([doi or URL or identifiers.org/DATABASE:ACCESSION])"

Response: I have included the following text in the main manuscript.

"This study includes no new experimental data deposited in external repositories. The datasets and computer code produced in this study are available in the following repository (GitHub): <https://github.com/mdtauqeer/activationNetwork>"

4) Code: The link for the codes for data analysis available in Github in the Data Availability section does not work. Please update the link to direct to the proper website. Please also be sure that a README file with practical use instructions for potential future users of your code is included in Github.

Response: The codes and associated datasets are now public. It is accessible to everyone. We have created a README file and explained the datasets and all the instructions are clearly mentioned.

5) Please rename "Competing Interest" to "Disclosure and competing interests statement". We updated our journal's competing interests policy in January 2022 and request authors to consider both actual and perceived competing interests. Please

review the policy <https://www.embopress.org/competing-interests> and update your competing interests if necessary.

Response: Competing Interest is renamed to 'Disclosure and competing interests statement'

6) Our journal encourages inclusion of *data citations in the reference list* to directly cite datasets that were re-used and obtained from public databases. Data citations in the article text are distinct from normal bibliographical citations and should directly link to the database records from which the data can be accessed. In the main text, data citations are formatted as follows: "Data ref: Smith et al, 2001" or "Data ref: NCBI Sequence Read Archive PRJNA342805, 2017". In the Reference list, data citations must be labeled with "[DATASET]". A data reference must provide the database name, accession number/identifiers and a resolvable link to the landing page from which the data can be accessed at the end of the reference. Further instructions are available at <https://www.embopress.org/page/journal/17574684/authorguide#referencesformat>.

Response: In our manuscript we have used 1) the computational model Yeast9, 2) enzyme activator information from BRENDA database, 3) gene expression dataset from Array Express database and 4) protein expression dataset from Messner et al. To appropriate positions we have cited these papers according to above instructions.

7) The "Methodology" section should be renamed to "Methods".

Response: I have corrected it.

8) All Materials and Methods need to be described in the main text using our 'Structured Methods' format. According to this format, the Methods section includes a Reagents and Tools Table (listing key reagents, experimental models, software and relevant equipment and including their sources and relevant identifiers) followed by a Methods and Protocols section describing the methods, ideally using a step-by-step protocol format. The aim is to facilitate adoption of the methodologies across labs.

Please download and fill our Reagents and Tools Table template (.docx), which you can find in our author

guidelines: <https://www.embopress.org/page/journal/14693178/authorguide#structured-methods>.

An example of a Method paper with Structured Methods can be found here: <https://www.embopress.org/doi/10.15252/msb.20178071>. "

Response: In our manuscript we have not used any experimental model of reagents. In the Reagent Table, I have added the computational model, list of tools, software and databases used for the analysis.

9) Please place individual sections of the manuscript in the following order: Title page - Abstract & Keywords - Introduction - Results - Discussion - Methods - Data Availability - Acknowledgements - Disclosure and Competing Interests Statement - References - Figure Legends - Expanded View Figure Legends.

Response: The manuscript is now re-structured according to your instructions.

10) For the figures and figure legends, please take care of the following:

- Please upload Figures EV1-5 as individual Figure files (like the main figures) with legends included in main manuscript file.
- Please make sure to update the callouts of all figures in the main manuscript text. Currently panel callouts are missing for Figure EV2 and Figure EV3.
- Please indicate the statistical test used for data analysis in the legends of figures 3B, 4D
- Please note that the box plots need to be defined in terms of minima, maxima, centre, bounds of box and whiskers, and percentile in the legends of figures 3B, 4A, B; EV3 A-C; EV4
- Please note that information related to n is missing in the legends of figures 3B, 4A, B; EV3 A-C; EV5

Response:

- Figures EV1-5 are uploaded as individual figures in TIFF format, and their legends are included in the manuscript.
- All Figures EV1-5 are cited in appropriate positions.
- Statistical tests were mentioned in relevant figures.
- Box plots are made as advised, and added the following text in legend "*Box plots display the median, the upper and lower quartiles, and the minimum and maximum values represented by the whiskers.*"
- In all relevant figures, n number has been added.

11) Dataset EV legend: The source file name, title and legend for Supplementary File 1 all need to be updated to Dataset EV1. This file should be uploaded individually as a Dataset file with the legend removed from the main manuscript file and included as a separate tab/sheet in the Excel file. Please also ensure that the callouts for this file are correct in the main manuscript.

Response: I have removed the legend for Dataset EV file from the main manuscript and mentioned it in a different sheet of the Dataset EV excel file. This dataset was cited in the main text to appropriate position.

12) Appendix file: Please upload the Appendix as a single PDF (no separate image files are needed).

Response: I do not have any appendix file. I am not sure which file you are referring to.

13) Synopsis:

- Synopsis image: Please provide a graphic that summarises the main findings of the manuscript on a glance and upload it as a high-resolution jpeg file 550 pixels wide x (300-600) pixels high.

- Synopsis text: Please provide a short standfirst (maximum of 300 characters, including space), limit the bullet points to max. 5 and upload it as a separate .doc file. Please write the bullet points to summarise the key NEW findings. They should be designed to be complementary to the abstract - i.e. not repeat the same text. We encourage inclusion of key acronyms and quantitative information (maximum of 30 words / bullet point). Please use the passive voice.

Response: The figure and text for synopsis is available as instructed.

14) As part of the EMBO Publications transparent editorial process initiative (see our policy here: https://www.embopress.org/transparent-process#Review_Process), Molecular Systems Biology will publish online a Peer Review File (PRF) to accompany accepted manuscripts. This file will be published in conjunction with your paper and will include the anonymous referee reports, your point-by-point response and all pertinent correspondence relating to the manuscript. Let us know whether you agree with the publication of the PRF and as here, if you want to remove or not any figures from it prior to publication. Please note that the Authors checklist will be published at the end of the PRF.

Response: Please publish Peer Review File. I have no objection to that.

15) After your paper is published, we will promote it on social media. If you have any handles or hashtags for Bluesky you would like included, please let us know.

Response: I have an account on platform X (formerly twitter, ID is **mdtauqeer**). Please let me know if you promote using twitter.

16) Please provide a point-by-point letter INCLUDING my comments and your detailed responses (as Word file).

Response: This file contains my point-by-point response to all of your comments.

24th Apr 2025

Manuscript number: MSB-2025-12861RR

Title: An enzyme activation network reveals extensive regulatory crosstalk between metabolic pathways

Dear Dr. Alam,

Thank you again for sending us your revised manuscript. We are now satisfied with the modifications made and I am pleased to inform you that your paper has been accepted for publication.

Yours sincerely,

Poonam Bheda, PhD
Scientific Editor
Molecular Systems Biology
